# Global genetic rewiring during compensatory evolution in the yeast polarity network

Enzo Kingma, Marieke Glazenburg, Karel Olavarria 🄳 & Liedewij Laan 🄳 ✉

## Abstract

**Functional defects resulting from deleterious mutations can often be restored during evolution by compensatory mutations. Importantly, this process can generate the genetic diversity seen in networks regulating the same biological function in different species. How the options for compensatory evolution depend on the molecular interactions underlying these functions is currently unclear. We investigate how gene deletions compensating for a defect in the polarity pathway of *Saccharomyces cerevisiae* impact the fitness landscape. Using a transposon mutagenesis screen, we demonstrate that gene disruption tolerance has changed on a genome-wide scale in the compensated strain. An analysis of the functional associations between the affected genes reveals that compensation impacts cellular processes that have no clear connection to cell polarity. Moreover, genes belonging to the same process tend to show the same direction of tolerance change, indicating that compensation rewires the fitness contribution of cellular processes rather than of individual genes. In conclusion, our results strongly suggest that functional overlap between modules and the interconnectedness of the molecular interaction network play major roles in mediating compensatory evolution.**

**Keywords** Compensatory Evolution; Transposon Mutagenesis; Cell Polarity; Modularity
**Subject Category** Evolution & Ecology

## Introduction

Cells must perform a series of complex tasks to stay alive and replicate. The successful execution of these tasks depends on a dense network of proteins, DNA, RNA, and other small molecules that physically interact. For various cellular functions, research in model organisms has unraveled the order and timescale of these interactions. Important examples of elucidated pathways are the endocytic (Burston et al, 2009; Kaksonen et al, 2005), cytokinesis (Pollard and O'Shaughnessy, 2019), and cell polarity pathways (Rappel and Edelstein-Keshet, 2017). Results obtained from model organisms are often extrapolated to other biological systems based on the assumption that the molecular mechanisms underlying the

same function are shared among different species. While molecular conservation has been demonstrated in several cases, it is becoming increasingly clear that components essential for a function in one species may be absent in another species that performs the same function (Diepeveen et al, 2018; Balasubramanian et al, 2004; van Hooff et al, 2017; Conibear, 2010). Thus, pathway composition is flexible during evolution, even when the same biological functionality is retained.

One of the possible mechanisms that can drive the emergence of different pathways performing the same cellular function is compensatory evolution. During compensatory evolution, the fitness effects of deleterious mutations, often in the form of gene loss or loss-of-function mutations, are alleviated by the acquisition of secondary mutations elsewhere in the genome. These secondary mutations effectively restore the perturbed pathway without reverting the original defect, thereby exploiting degenerate mechanisms to perform the same function. How the architecture of the molecular interaction network within the cell provides opportunities for genetic compensation of functional defects is unclear (Szamecz et al, 2014; Blank et al, 2014).

A prominent view is that molecular interaction networks are organized into dynamic modules (Hartwell et al, 1999; Han et al, 2004). Here, a module regulates a single physiological function and consists of a group of proteins that interact extensively with each other, but sparsely with the rest of the network. Proteins may belong to different modules at different points in time, making module composition dynamic (Han et al, 2004; Chang et al, 2013). Importantly, the modular organization of the interaction network has been reasoned to improve cellular evolvability (Hartwell et al, 1999; Wagner et al, 2007). Constructing a larger network from smaller, modular building blocks that interact sparsely would limit the pleiotropic effects of mutations, allowing each function to be optimized independently during evolution. In line with this role for modularity during adaptation, laboratory evolution experiments have shown that compensatory mutations preferentially arise in genes that are functionally related to the gene that was initially perturbed (Szamecz et al, 2014; Rojas Echenique et al, 2019). However, for modules to truly evolve independently, compensatory mutations must not alter the genotype-fitness relationship of other modules. Whether the effects of compensatory mutations on the genotype-fitness landscape are indeed only local is currently unexplored. In addition, doubts have arisen regarding the relevance and benefits of modules in network evolution due to findings from models, computer simulations, and phylogenetic analyses (Kashtan et al, 2009; ten Tusscher and Hogeweg, 2011; Snel and Huynen, 2004).

Department of Bionanoscience, Kavli Institute of Nanoscience Delft, Delft University of Technology, Delft 2629 HZ, The Netherlands. ✉E-mail: L.Laan@tudelft.nl

An appealing model system to study compensatory evolution is the comparatively well-studied cell polarization pathway (Styles et al, 2013; Altschuler et al, 2008; Goryachev and Leda, 2017). Cell polarization remains functionally conserved across the diverse branches of the tree of life (Thompson, 2013; Etienne-Manneville, 2004; Nelson, 2003). However, components essential for polarization in one species can lack orthologs in other species (Diepeveen et al, 2018), suggesting that compensatory evolution occurs naturally within this pathway. As in most eukaryotes, cell polarization in *Saccharomyces cerevisiae* is regulated by the cycling of the protein Cdc42 between a cytosolic inactive and a membrane-bound active state. Extensive research has uncovered the following three interconnected functional modules that collectively form the polarity pathway (Marco et al, 2007; Woods and Lew, 2019; Casamayor and Snyder, 2002; Freisinger et al, 2013): actin-based transport, spatial cues and a reaction-diffusion module (Fig. 1A). While all three modules contribute to cell polarization, the reaction-diffusion module has been identified as its primary driver in wild-type cells. This functional module critically depends on the protein Bem1 (Irazoqui et al, 2003; Klünder et al, 2013), which serves as a scaffold that connects most other components of the module. Deletion of Bem1 is highly disruptive and nearly abolishes the ability of cells to polarize (Kozubowski et al, 2008; Laan et al, 2015). Intriguingly, an earlier study has shown that this defect can be almost completely compensated by the additional deletion of two other proteins, Bem3 and Nrp1 (Fig. 1B, Laan et al, 2015). Bem3 is a known component of the reaction-diffusion module (Zheng et al, 1994; Knaus et al, 2007), while Nrp1 is a protein of unknown function for which a role in polarity establishment might still be uncovered.

In this study, we evaluate how compensatory gene deletions within the cell polarity pathway influence the global fitness landscape and explore the role of modules in shaping evolutionary dynamics. We utilize a transposon mutagenesis screen to determine changes in gene disruption tolerance and demonstrate that compensatory evolution affects the fitness landscape on a genome-wide scale (Fig. 1C). Despite compensation being achieved solely by gene deletions, the number of genes with increased and decreased insertion tolerance in the compensated strain is nearly equal.

This finding indicates that evolution through gene loss does not always reduce mutational robustness. Because the evolutionary dynamics of proteins are often linked to their topological role in the protein interaction network (Fraser et al, 2003; Alvarez-Ponce et al, 2017), we attempted to explain our observed pattern of disruption tolerance changes using the structure of the protein-protein interaction network of *S. cerevisiae*. However, we were unable to find any meaningful correlation with the protein network structure. Instead, studying the functional associations among the affected genes revealed that genes related to the same process tend to share the same direction of insertion tolerance change. Thus, although changes in gene disruption tolerance do not remain isolated to the originally perturbed module, functionally related genes appear to respond similarly during adaptation. Cell polarity was amongst the biological processes enriched for genes with a decreased tolerance, suggesting that redundancies between different modules within the polarity pathway mediate the capacity to compensate for gene loss. Collectively, our data demonstrates that mutations compensating for defects in a single functional module can lead to genome-wide changes in the gene–fitness relationship.

# Results and discussion

## Compensatory mutations cause global changes in the fitness landscape

The unperturbed wild-type strain and the compensated *bem1Δ-bem3Δnrp1Δ* mutant, which we refer to as the polarity mutant, have similar fitness under laboratory conditions (Laan et al, 2015). However, whether the recovery of fitness after the loss of Bem1 is accompanied by restoration of the original fitness landscape has not been determined. The extent to which the fitness landscape is affected will reflect the molecular mechanism through which the gene deletions provide compensation. We considered the following two mechanisms for compensation to be likely.

The first mechanism relates to the argument that gene loss events during evolution often occur as a form of redundancy reduction (Albalat and Cañestro, 2016). Similarly, modules can be redundant because they have largely overlapping functions. In the case of the polarity mutant, this implies that the reaction-diffusion module is redundant with one or more other modules that can drive polarity establishment. Deleting Bem1 causes the reaction-diffusion module to malfunction and interfere with these redundant modules. Consequently, further inactivation of the reaction-diffusion module through the deletion of *BEM3* and *NRP1* becomes advantageous. A recent study introduced a mathematical model that explains how functional redundancies between modules could be leveraged to form a latent pathway for polarity establishment (Brauns et al, 2023). If compensation depends solely on the activation of these redundant mechanisms, we expect the fitness landscape to remain largely unchanged, with the exception of the decreased dispensability of the modules that are part of the latent pathway.

The second mechanism depends on the compensatory gene deletions facilitating novel interactions between other, non-mutated, proteins. According to the structure-function paradigm, a protein adopts a unique three-dimensional structure that ultimately determines its function. However, advancements in structural biology have revealed that protein structures are significantly more dynamic than previously believed (Nussinov et al, 2023). For example, up to 45% of residues in the eukaryotic proteome are found in intrinsically disordered regions of proteins (Xue et al, 2012), which lack a well-defined structure. Importantly, this structural flexibility enables protein function to adapt in response to varying cellular contexts and binding partner availability (Miskei et al, 2017). Thus, the deletion of Bem3 and Nrp1 could compensate for the loss of Bem1 by modifying the functional properties of other proteins in the proteome, potentially leading to extensive rewiring of the interaction network. We anticipate that such rewiring of the protein-protein interaction network will result in global changes to the fitness landscape.

To uncover differences in gene disruption tolerance between the wild-type strain and the polarity mutant on a genome-wide scale, we performed the transposon mutagenesis screen SATAY (Michel et al, 2017). This screen utilizes the MiniDS transposon to generate

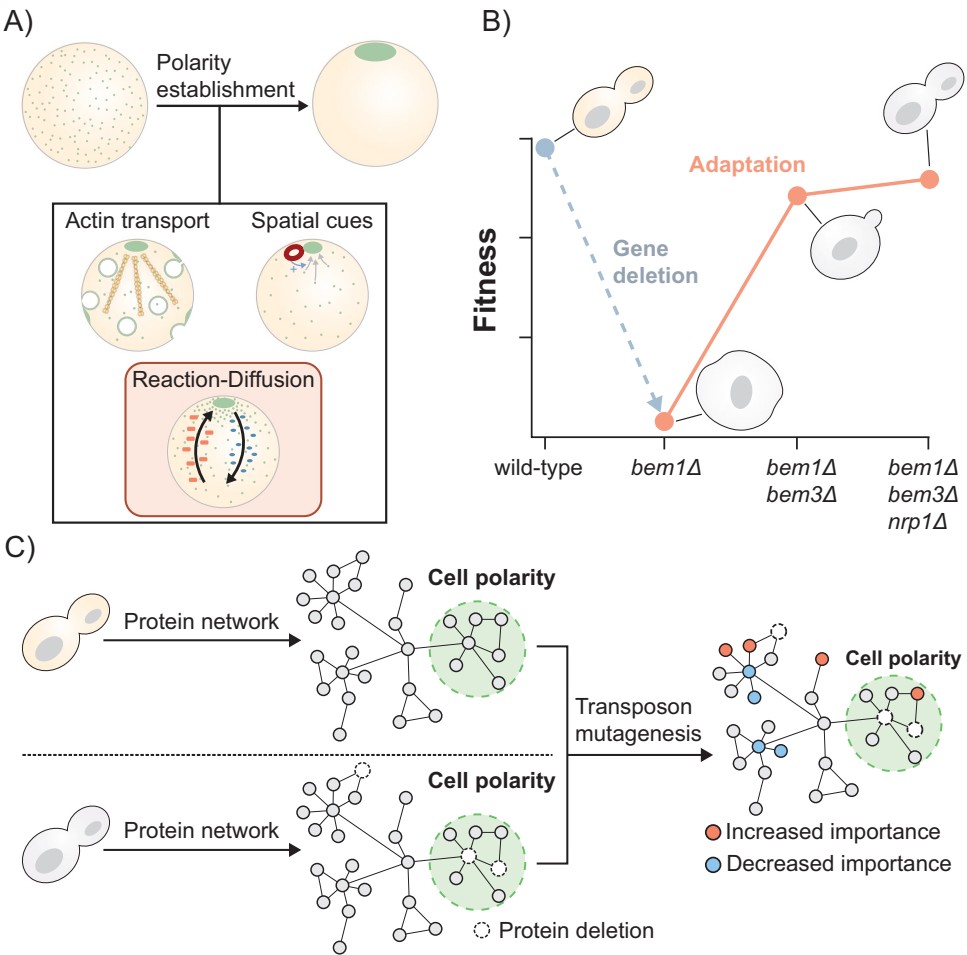

**Figure 1. Determining the consequences of compensatory evolution of the polarity pathway on global cellular physiology.**

(A) Polarity establishment is an evolutionary conserved function of nearly all cell types. In budding yeast, cell polarization is initiated by the formation of an asymmetric distribution of Cdc42 (shown in green). Actin-based transport, spatial cues, and a Bem1-mediated reaction-diffusion module have all been suggested to be involved in Cdc42 polarization dynamics. In this study, we examine how the fitness landscape changes after evolutionary compensation for a genetic perturbation in the reaction-diffusion module. (B) The phenotypic effects resulting from the deletion of the polarity protein Bem1 can be compensated by the additional loss of Bem3 and Nrp1. The resulting polarity mutant has a fitness similar to a wild-type strain under standard laboratory conditions. The figure is based on previous data obtained by Laan et al (2015). (C) We use a transposon mutagenesis screen to determine the effects of compensatory evolution on the genotype-fitness relationship on a global scale. Genes that differ in disruption tolerance between the wild-type strain and the polarity mutant are anticipated to encode proteins with modified physiological roles.

a library of gene disruption mutants. In short, a plasmid carrying the MiniDS transposon is transformed into the cell. By adding galactose to the growth media, the transposon is induced to migrate from the plasmid to a random location within the genome. This genomic integration disrupts the original sequence, typically abolishing gene function when it occurs within a coding region. By performing this process in a population of cells, a library of gene disruption mutants is created, with each mutant carrying a transposon at a different genomic location. The mutant library is subjected to a pooled fitness assay immediately after its generation. Subsequently, the genomic DNA is extracted from the mixed population and sequenced. Here, the genomic regions flanking the transposon insertion site serve as tags to distinguish the different mutants. The fitness of a mutant (relative to the bulk fitness of the mutant library) is reflected by the number of reads that align to the corresponding insertion site.

We generated six SATAY libraries of the wild-type strain and six libraries of the polarity mutant (Fig. 2A). Overall gene disruption tolerance was estimated by summing all read counts that map within a particular coding region. We note that these summed read counts are independent of the fitness differences between the polarity mutant and the wild-type strain, but rather depend only on the average fitness of the other transposon mutants present in the library. To compare datasets from independent experiments, the read count distributions were normalized (see Appendix Section A) using a Beta-geometric normalization (Appendix Fig. S1a,b), followed by a median of ratios normalization (Appendix Fig. S1c). From the six replicates of each genetic background, we determined the mean ($\mu$) and variance ($\sigma$) of the summed read count for each gene (Fig. 2A). The changes in the mean total read counts between the wild-type strain and polarity mutant (Dataset EV1) are presented in the form of a volcano plot in Fig. 2B. Interestingly,

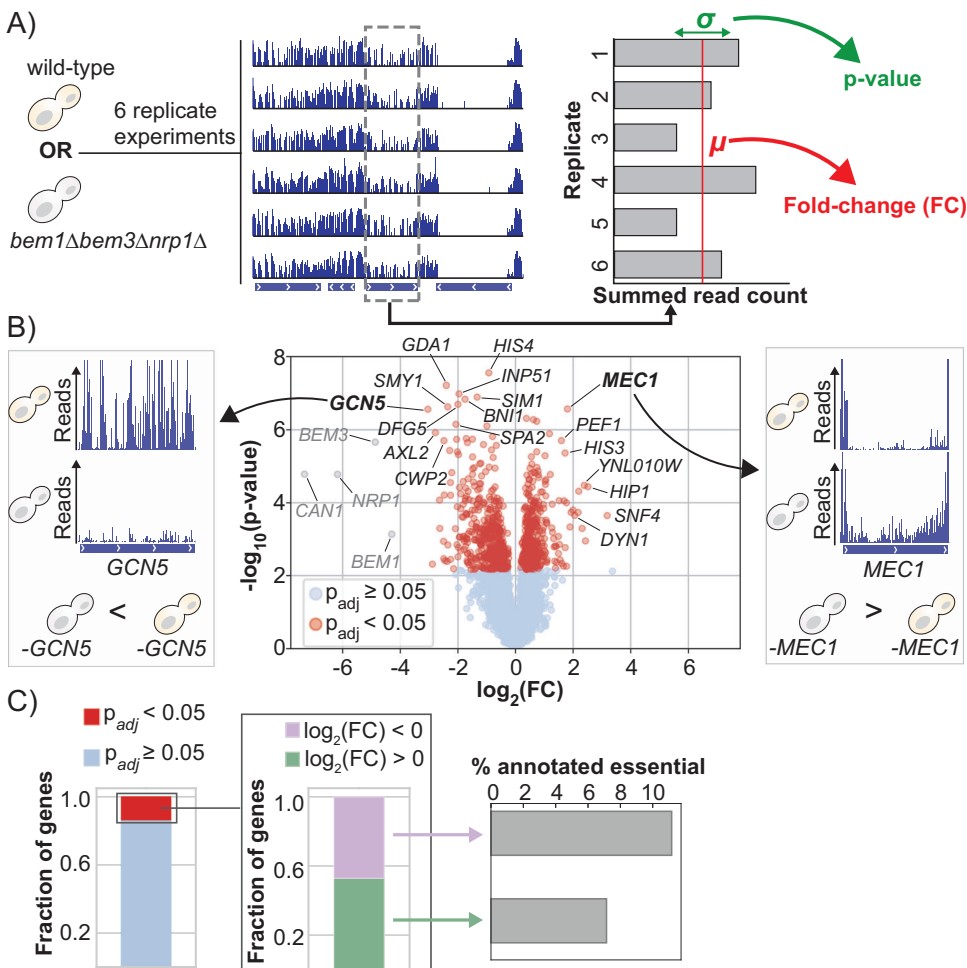

**Figure 2. Compensatory gene deletions change the gene–fitness relationship on a genome-wide scale.**

**(A)** We created six replicate SATAY libraries (biological replicates) of the wild-type strain and six replicate libraries of the polarity mutant. To estimate disruption tolerance, we summed the read counts of transposon insertions that map to the same coding region. The mean $\mu$ and variance $\sigma$ of the total read count for each gene were calculated from six replicate libraries for each genetic background. The mean and variance were subsequently used to determine the log-fold change and significance of the difference in total read count between the two genetic backgrounds. **(B)** Volcano plot of the changes in gene disruption tolerance showing the log-fold change (FC) and the significance value of the differences in disruption tolerance between the two strains. Positive log-fold changes relate to genes that have a higher disruption tolerance in the polarity mutant relative to the wild-type strain. Statistical significance was determined with Welch's $t$ test and corrected for multiple hypothesis testing with the Benjamini–Hochberg procedure. Genes that have been grayed out correspond to genes that were synthetically deleted from the genome of the polarity mutant during strain construction. **(C)** Approximately 13% of all annotated genes display a significantly altered tolerance to transposon disruptions between the two strains. An analysis of the sign of gene disruption tolerance changes in this set of genes shows that the ratio of genes that increase their relative fitness and those that decrease their relative fitness in the polarity mutant is nearly equal. Both the set of genes with increased fitness and the set of genes with decreased fitness include several genes that have been annotated as essential. Source data are available online for this figure.

we find that a large fraction of the annotated genes in the genome have a significant difference in disruption tolerance between the wild-type strain and the polarity mutant. Specifically, a total of 883 genes, which constitute more than 13% of all annotated genes in the yeast genome, have an altered tolerance (Fig. 2C). Preliminary data indicate that genes with altered disruption tolerance indeed have distinct fitness effects in the wild-type and polarity mutant backgrounds (Appendix Section B and Appendix Fig. S4). As a reference, a comparison of SATAY libraries obtained from the same genetic background revealed no statistically significant differences (Fig. EV1B,C).

The percentage of genes with an altered disruption tolerance observed between the two closely related genetic backgrounds in

our study is comparable to the proportion previously reported for genes exhibiting background-dependent fitness effects across distinct natural isolates (Caudal et al, 2022; Galardini et al, 2019). Considering that natural isolates will typically display greater genomic divergence than the two strains analyzed here, we anticipated finding a smaller number of genes with a background-dependent disruption tolerance. One difference is that our analysis method detects changes in the fitness effect of gene disruptions on a continuous scale, while the earlier reports focused on a binary classification of genes as essential or non-essential. As a result, our analysis captures differences in fitness effects that would likely have remained undetected by approaches limited to such a binary classification of genes. Despite the comparatively higher

sensitivity of our method to detect fitness differences, we likely still underestimate the number of genes that have a background-dependent disruption tolerance. Notably, by summing read counts across the entire open reading frame, we neglect possible domain effects. Furthermore, by restricting our analysis to coding regions, we are unable to capture differences in the fitness effects caused by changes in gene expression.

The impact of compensatory mutations on the fitness landscape may vary depending on the environment in which fitness is assessed. (Filteau et al, 2015; Mullis et al, 2018). In a recent survey of changes in gene essentiality between natural isolates of *S. cerevisiae*, it was proposed that the majority of the observed variation was due to environmental factors (Caudal et al, 2022). That is, it was argued that most genes that emerged as differentially essential in their screen were caused by strain-specific responses to growth in non-standard media. The mutagenesis screen we performed has a similar limitation. SATAY requires transposition to be induced in media in which galactose is the sole carbon source (Michel et al, 2017), while the phenotypic similarity between our two strains has only been verified for growth under standard conditions (Laan et al, 2015). We therefore cannot exclude that we would have observed less genes with a differential disruption tolerance if we had performed a screen in a different environment. However, we expect that the effect of the environment on our results is limited for the following two reasons. First, the overall genetic divergence between our strains should be smaller than what is typical for natural isolates that may have adopted different lifestyles to adapt to their niche. A different study indeed showed that most genetic interactions are conserved across environments (Costanzo et al, 2021). Second, while we did identify a cluster enriched for metabolically related genes in our gene set (which is possibly an artifact of the method used for strain construction, see Appendix Section C), genes involved in galactose metabolism or respiration are not overrepresented in this cluster.

Overall, these data show that compensatory evolution affects a substantial fraction of the genome. Despite the intention to limit the initial perturbation to a single functional module (Laan et al, 2015), the effects of compensation on the fitness landscape extend far beyond this module. These widespread changes in gene disruption tolerance suggest that the recovery of cell polarity cannot be accomplished by merely reallocating the lost functions of the disrupted module to other redundant modules. However, while many of the genes with an altered tolerance lack a clear relation to cell polarity, the set also includes several genes that are known to encode for polarity regulators (for example, *SPA2*, *BNI1*, and *AXL2*). Functional redundancies between modules may therefore still mediate compensation, but seemingly not without affecting the disruption sensitivity of genes regulating other cellular processes.

## Compensatory evolution results in both increased and decreased disruption tolerance

To better understand how compensatory evolution reshapes the gene–fitness landscape, we examined the direction of disruption tolerance changes. We find that the set of genes with increased tolerance (414 genes) is nearly as large as the set of genes with decreased tolerance (468 genes). Moreover, both sets include several genes that have been annotated as essential (Fig. 2C). Interestingly, the average log-fold change in disruption tolerance is greater (*P* value =

9.7e-29, Wilcoxon rank-sum test) for genes with decreased tolerance (mean = −1) than for those with increased tolerance (mean = 0.66). This asymmetry in the fold-change distribution (Fig. EV1A) suggests that, while the polarity mutant may not rely on more genes for fitness compared to the wild-type strain, the fitness impact of disrupting a gene on which it has become dependent after compensatory evolution is more pronounced. However, since the magnitude of the fold change also depends on the size of the affected region relative to the total gene length, it is possible that genes with increased disruption tolerance more often encode for multi-domain proteins in which only one domain exhibits a variation in disruption tolerance.

While the polarity mutant and wild-type strains are phenotypically similar, the mutant exhibits a modest but statistically significant reduction in maximum growth rate (Laan et al, 2015). This difference in relative fitness raises the possibility that diminishing returns or increasing cost epistasis- in which the fitness effect of a mutation depends on the fitness of the genetic background in which it occurs (Kryazhimskiy et al, 2014)—contributes to the observed differences in gene disruption tolerance. Such an effect could introduce a systematic bias toward identifying a greater number of genes with an altered disruption tolerance in the polarity mutant. Importantly, these instances would not indicate genuine differences in genetic architecture between the two strains but rather arise from comparing beneficial and deleterious mutations of equivalent magnitude in genetic backgrounds with differing baseline fitness. However, we argue that global epistasis alone cannot fully explain the observed disruption-tolerance differences between the two strains for the following reasons. First, most mutations are expected to impair fitness in a wild-type genetic background (Venkataram et al, 2016; Baryshnikova et al, 2010). If increased cost epistasis were the primary driver of altered disruption tolerance between the two genetic backgrounds, we would expect a bias toward genes with a reduced disruption tolerance in the less-fit mutant strain. Second, genes annotated as essential and predicted to reduce fitness in the wild-type are found in both the gene sets with an increased and a decreased disruption tolerance in the polarity mutant (Fig. 2C), indicating that essentiality does not determine the direction of tolerance change.

It is surprising that the number of genes with increased and decreased insertion tolerance is nearly balanced. Intuitively, compensatory deletions would be expected to mediate pathway restoration by shifting dependencies to alternative cellular components that take over the lost functions. Our finding that many genes also display an increased disruption tolerance in the compensated strain indicates that the compensatory gene deletions reorganize the gene–fitness relationship rather than simply redistribute the lost functions to other, possibly redundant, genes.

An important question is to what degree the individual compensatory mutations drive the global genetic rewiring that we observe. For instance, the deletion of Bem3, which plays a crucial role in polarity establishment, may have a localized impact on the fitness landscape, whereas the loss of the RNA-binding protein Nrp1 could trigger more extensive genetic changes on a global scale. A comparison with the genetic interactions mapped by a genome-wide SGA screen reveals that only a small subset (<30%) of the digenic interactions involving *BEM1*, *BEM3*, and *NRP1* appear as differentially tolerant in our SATAY dataset (Fig. EV2). This finding suggests the prevalence of higher-order epistasis during compensatory evolution and indicates that global rewiring is not driven by the additive effects of digenic interactions. Notably, while

the signs of the genetic interactions for *BEM3* and *NRP1* show limited agreement between the SGA screen and SATAY (Fig. EV2B,C), those for *BEM1* are relatively well-preserved (Fig. EV2A). This suggests that the genetic interactions associated with the strong initial perturbation are better retained compared to those of compensatory mutations.

## Changes in gene disruption tolerance cannot be predicted from the structure of the protein-protein interaction network

Are all genes equally likely to have a modified fitness in the compensated strain? If not, establishing a relationship between gene characteristics and features of the molecular interaction network could provide valuable insights for making evolutionary predictions. Earlier studies have indicated a correlation between a protein's evolutionary rate and its position within the protein interaction network (Alvarez-Ponce et al, 2017). For example, highly connected proteins (hubs) are more likely to be essential (Jeong et al, 2001) and tend to evolve slower (Fraser et al, 2003; Fraser et al, 2002) than less-connected proteins, possibly because they more frequently participate in important interactions (Batada et al, 2006; He and Zhang, 2006). These findings suggest that the topology of protein interaction networks provides relevant information to understand the evolutionary dynamics of individual proteins.

We constructed a protein-protein interaction network (PPI) to see if we could identify a correlation between the topology of this network and the changes in gene disruption tolerance in the compensated strain. Protein interactions were taken from the BioGrid database (Stark, 2006; Oughtred et al, 2021). Because the ability to derive a correlation between evolutionary dynamics and PPI topology depends on network quality, we only included interactions from the Multi-Validated dataset (see "Methods"). The resulting PPI network consists of 3799 proteins (nodes) and 17,205 undirected interactions (edges). While the majority of the proteins are part of a single, fully connected network, a small portion is disconnected from the main graph (Fig. 3A). In addition, the network represents only 65% of the estimated 5800 proteins in the yeast proteome and is therefore incomplete. For the remaining 35% of the proteome, the protein interaction pattern has not been validated by at least two independent studies or experimental methods. Among the absent proteins is Nrp1, which is one of the proteins that provides compensation for the loss of Bem1 when deleted. Despite these limitations, we successfully validated that proteins in our network with more interactions are more likely to be essential (Fig. 3B,C), a relation formally referred to as the centrality-lethality rule (Jeong et al, 2001)). This demonstrates that crucial structural features of the network can still be inferred.

The importance of a node for different topological features of a network can be described using network centralities (Koutrouli et al, 2020). Here, we focus on three commonly used centralities: degree, betweenness, and closeness. These centralities relate to the role of a protein in the PPI network. For example, hub proteins at the center of a module that interact with many other proteins have a high degree, while proteins that mediate connections between modules have a high betweenness. We provide an overview of the distributions of these centralities in our network in Fig. EV3B–D. Importantly, Fig. EV3B shows that the degree distribution follows

the sub-linear dependency characteristic of biological networks when plotted on a log-log scale. Furthermore, we show for 4 polarity proteins that those which strongly impact cellular viability (Cdc42 and Bem1) can clearly be distinguished from those which have a lesser effect (Bem3 and Bem2) based on their degree and betweenness (Fig. EV3B,D). The relationship with closeness is, however, less clear (Fig. EV3C).

Next, we attempted to derive a correlation between the three centrality measures described above and the likelihood that gene disruption tolerance has changed after compensatory evolution. However, the likelihood distributions are uniform for all centralities (Fig. 3D–F), indicating a lack of correlation. Thus, the genes that will have a differential fitness effect between the two strains cannot be inferred from these three centralities. While we are able to validate the presence of several features that are typically observed in PPI networks, we cannot reject the possibility that the lack of correlation is a consequence of the poor quality of our PPI network. The accuracy of annotated physical interactions is known to substantially vary between datasets (von Mering et al, 2002; Gillis et al, 2014). To exclude interactions with low confidence, we used the Multi-Validated dataset available from the BioGRID to construct our PPI network. However, network quality does not only depend on the confidence that an interaction exists, but also on the confidence that an interaction is absent (van Dam and Snel, 2008). In addition, bias is a significant problem in the mapping of interaction networks, as weak and transient interactions are often discarded because they cannot be captured with high confidence by affinity-based screens. Such limitations may have contributed to the lack of correlation we observed between a protein's position within the PPI network and the probability that its corresponding gene exhibits altered disruption tolerance.

The lack of a relationship between changes in disruption tolerance and PPI network structure obscures a mechanistic explanation for how the compensatory gene deletions restore fitness. In particular, the observation that many genes appear to become increasingly dispensable after these compensatory deletions (section B) is counterintuitive. A mechanistic explanation for how gene loss during evolution can increase the dispensability of other genes has been proposed previously. Specifically, when the loss of a protein fully inactivates an associated pathway, the additional loss of other proteins in the same pathway does not incur additional fitness costs (Domingo et al, 2019; van Leeuwen et al, 2017). In this way, the loss of a protein can appear to buffer mutations in other proteins. However, this explanation does not apply when gene deletions restore the function of a perturbed pathway. Although our results do not provide direct evidence, we propose that gene loss may trigger widespread rewiring of the physical interaction network based on the following reasoning. The fitness effect of a gene deletion is linked to the physiological role of its encoded protein. However, it is increasingly recognized that proteins are not static entities but exist as structural ensembles (Nussinov et al, 2019; Nussinov et al, 2023). The preferred conformation depends on the environment, making protein function highly context dependent (Schwille and Frohn, 2022). We speculate that in a similar manner, protein structure may also be influenced by the presence or absence of specific binding partners (Boehr et al, 2009; Miskei et al, 2017). The loss of one binding partner could therefore induce conformational changes that alter a protein's affinities for alternative partners. In this way, the deletion of one protein can

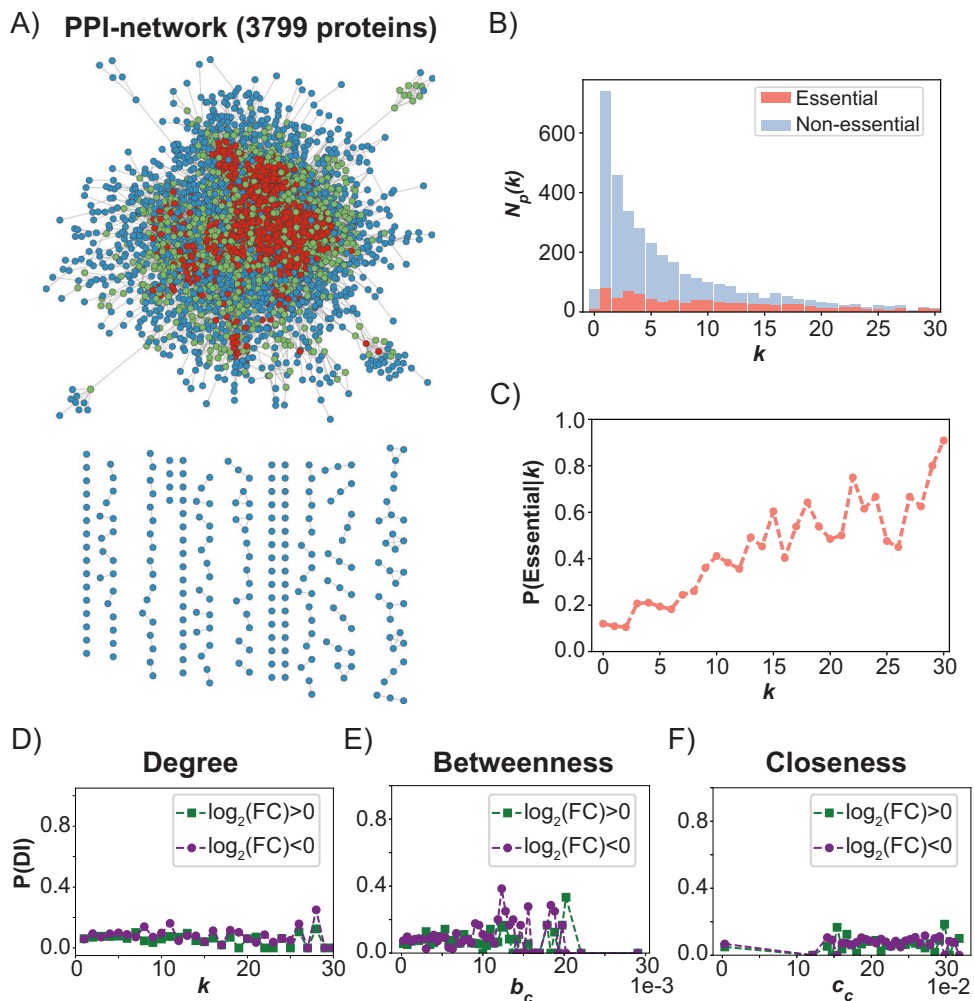

**Figure 3. The likelihood of gene disruption tolerance change does not correlate with the centralities of the protein-protein interaction (PPI) network.**

(**A**) Visualization of the PPI network constructed from the interactions annotated by the BioGRID database. Nodes are colored according to their degree $k$ (blue: $k \leq 3$, green: $4 \leq k \leq 10$, red: $k > 10$). (**B**) Stacked histogram of the number of proteins $N_p(k)$ with degree $k$. Proteins for which the gene is annotated as essential are shown in red, non-essential proteins are shown in blue. The histogram shows that the distribution for $k$ is roughly uniform for essential proteins, while it is skewed toward lower values of $k$ for non-essential proteins. (**C**) The probability that a node is essential as a function of $k$. The graph displays an increasing trend for the probability that a node is essential for larger values of $k$. Combined with the histogram shown in (**B**), it can be seen that this increasing trend is due to an enrichment of essential proteins among those with a high degree, and not because most essential proteins have a high degree. (**D–F**) The conditional likelihoods for changes in gene disruption tolerance as a function of the degree (**D**), betweenness (**E**) and closeness (**F**) centralities of their corresponding protein in the network shown in (**A**). For the three centralities shown in (**D–F**), the likelihood distribution is approximately uniform.

initiate a cascade of structural and functional adjustments that lead to large-scale rewiring of the protein interaction network.

## Genes with an altered disruption tolerance are associated with a diverse array of cellular processes

The lack of a correlation between PPI network structure and the likelihood of observing a change in gene disruption tolerance prompted us to see if our dataset could perhaps be structured based on a different feature. Importantly, the lack of dependence on the PPI network does not exclude the possibility that genes participate in the same cellular process, as this does not require proteins to physically interact. These indirect functional relationships between genes can be captured with functional association networks.

We used the STRING database to create a functional association network from the genes with a differential fitness between the two genetic backgrounds (Szklarczyk et al, 2021; Doncheva et al, 2019; Szklarczyk et al, 2019). In this database, functional relations between genes are inferred by integrating information from multiple sources, such as text mining, molecular complex annotations, and (predicted) physical or genetic interactions. Each interaction is scored based on the quality of the evidence, allowing more weight to be given to high-confidence interactions.

Similar to our PPI network, the functional associations resulted in most genes being part of a single, connected, network from which only few nodes are disconnected (Fig. 4A). Analyzing the degree distribution indicated a scale-free structure of the network, suggesting it can be partitioned into smaller sub-graphs. We ran the

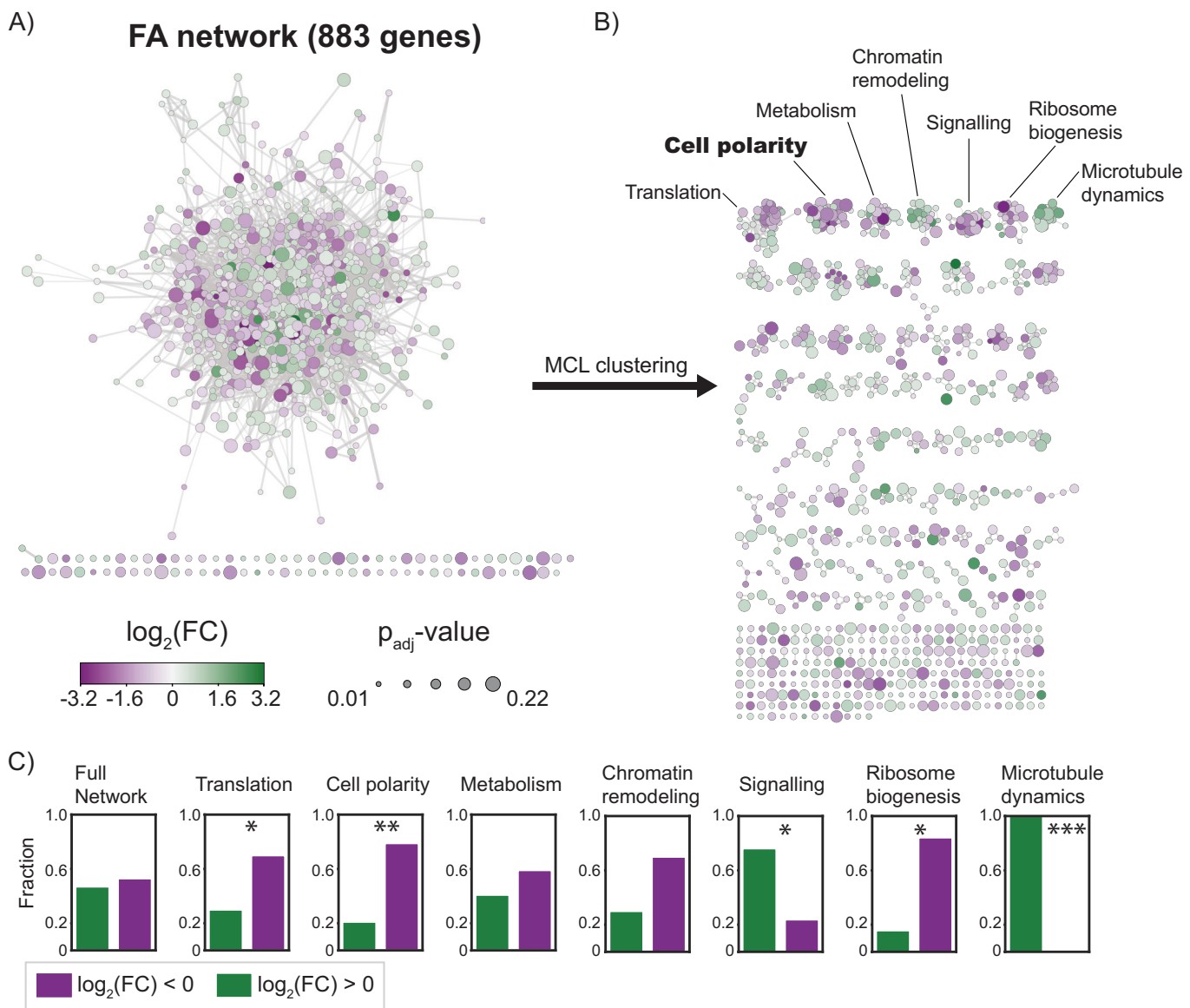

**Figure 4. Clustering of a functional association (FA) network reveals diversity of biological processes affected by compensatory evolution.**

(A) Graph of the FA network constructed from genes that have a differential fitness between the wild-type strain and the polarity mutant. Node color scales with the magnitude and sign of the fitness effect, and node size scales according to the significance level (adjusted *P* value). The graph shows that most genes are contained in a single, densely connected network. (B) The clusters formed from the graph shown in (A) by the Markov clustering algorithm (MCL). The clusters are arranged from large (top left) to small (bottom right). The biological process gene ontology enrichment is shown for the seven largest clusters. (C) Distribution of genes with a decreased (purple) and increased (green) fitness in the polarity mutant for the seven largest clusters. The clusters enriched for translation ($P = 1.3e - 2$), cell polarity ($P = 1.2e - 3$), and ribosome biogenesis ($P = 8.7e - 3$) are enriched for genes with decreased fitness. Conversely, the signaling ($P = 7.6e - 3$) and microtubule dynamics ($P = 1.1e - 5$) clusters are enriched for genes with increased fitness. *P* values are indicated in the plot by asterisks: *$P < 0.05$, **$P < 0.005$, ***$P < 5 \times 10^{-4}$. The significance level was determined with Fisher's exact test.

Markov Cluster (MCL) algorithm (van Dongen, 2008) to uncover these sub-graphs, assigning interaction confidence scores as weights. This approach uncovered 120 clusters, of which 49 consisted of three or more genes. For the seven largest clusters, we performed a gene ontology (GO) enrichment test for biological process GO terms (Figs. 4B and EV4). All seven clusters were significantly enriched for at least one GO term. Five clusters were enriched for the following terms related to cellular homeostasis: translation, signaling, chromatin remodeling, ribosome biogenesis,

and metabolism. However, the presence of a metabolic cluster is likely linked to the auxotrophic *HIS3* marker introduced into the polarity mutant during strain construction (see Appendix Section C and Appendix Fig. S5). The remaining two clusters were associated with components of the cell division machinery: cell polarity and microtubule dynamics.

When assessing how genes with increased and decreased fitness are distributed among the clusters, we found that genes within the same cluster were often, but not always, more likely to share the

same sign in fitness change than would be expected from a random partitioning (Fig. 4C). Specifically, the clusters enriched for genes related to translation, ribosome biogenesis and cell polarity were also enriched for genes with a decreased disruption tolerance. Alternatively, genes in the signaling and microtubule-enriched clusters were more likely to have an increased disruption tolerance.

Our analysis reveals widespread functional connections between the genes in our dataset. At the same time, by partitioning their functional association network into sub-graphs, we were able to identify that these genes are involved in a diverse set of cellular processes. Intriguingly, the changes in fitness dependency seem to occur at the cellular process level rather than being attributed solely to individual genes. This uniform response of genes regulating the same process resembles the observed monochromatic behavior of genetic interactions between modules of the yeast metabolic network (Segrè et al, 2005) and more generally between genes involved in the same pathway or complex (Costanzo et al, 2010).

When comparing the cellular processes that are important for survival between the polarity mutant and wild-type strain, we observe several notable differences. First, the general trend of genes involved in translation showing a reduced disruption tolerance in the polarity mutant suggests that this strain is more sensitive to variations in protein copy numbers. This consequence of the compensatory mutations was also predicted by the mathematical model of Brauns et al (2023). Second, the finding that genes regulating microtubule dynamics become increasingly dispensable is surprising, given that microtubules are required to perform vital cellular functions such as spindle positioning and nuclear migration (Carminati and Stearns, 1997; Boldogh et al, 2001). Thus, compensatory evolution appears to profoundly affect the genetic wiring of the cell, changing the patterns of essentiality and dispensability beyond the genes related to the perturbed module.

The observation that genes with differential insertion tolerance form functionally enriched clusters suggests that some level of modular organization exists within biological networks. This view is further supported by the finding that several of these clusters are enriched for either genes with increased or decreased fitness. At the same time, our data demonstrates that the compensatory evolution of one module of the polarity pathway affects the disruption tolerance of genes in many other modules, including those that have no clear relation to cell polarization. These findings complement results from an earlier study by Harcombe et al (2009), which showed that gene loss can be compensated for by mutations in functionally unrelated genes. Our results further demonstrate that, in addition to the compensatory mutations themselves, gene fitness across multiple cellular processes can change during compensatory evolution. It is conceivable that these changes in gene fitness will affect the further evolution of these modules.

## Compensatory evolution is mediated by redundancies within the polarity pathway

The wild-type strain and polarity mutant are nearly identical with respect to their polarization efficiency (Laan et al, 2015). The ability to polarize efficiently despite the loss of three genes implies the existence of an alternative pathway that is active in the absence of the reaction-diffusion module. Because it can function without adding any components, this pathway is considered to be latent or

hidden within the interaction network of the wild-type strain (Brauns et al, 2023). While reconstruction of the polarity mutant has demonstrated the existence of this latent pathway, its specific structure remains to be determined. In the previous section, we found a cluster enriched with polarity genes amongst those affected by the mutations compensating for defects in the reaction-diffusion module. The presence of this cluster implies that functional redundancies between the different modules of the polarity pathway could play an important role during compensatory evolution.

To uncover how these redundancies can be exploited to constitute an alternative polarization mechanism, we annotated the genes in the cell polarity cluster according to their gene ontology (Fig. 5A; Dataset EV2). This revealed that the majority of the genes encode for proteins that are part of one of the following three cellular components: (1) cortical actin patches, (2) the bud scar, and (3) the polarisome. Interestingly, each of these components relates to a different module of the known polarity pathway. Specifically, the bud scar provides a spatial cue that directs the axis of polarity, while cortical actin patches and the polarisome regulate the transport of active Cdc42 via endo- and exocytic vesicles, respectively. For all three cellular components, a substantial fraction (>20%) of the complete set of annotated genes is represented in our dataset.

While all three cellular processes identified above are active in wild-type cells, their contribution to cell polarization remains inconspicuous when the Bem1-dependent reaction-diffusion module is active. This feature has been attributed to the hierarchical relationship that normally exists between modules in the polarity pathway (Daalman et al, 2020). To determine if this hierarchy still exists within the latent pathway, we used the log-fold change distributions of genes belonging to each of the three modules. Specifically, the module that takes on the most prominent role in the latent pathway should show the greatest change in gene disruption tolerance. However, quantitatively comparing log-fold changes in read counts summed on the gene level can give distorted results because their values depend on gene size. For example, if the region within a gene that exhibits a differential tolerance to transposon disruptions is small relative to the total gene size, the log-fold change value will be attenuated. To address this issue, we performed a more refined estimate of changes in gene disruption tolerance by using a moving average with a 300 bp window over the genes to obtain a fitness profile across the coding region (Fig. 5B). The choice for a window size of 300 bp was made based on the average size of a protein domain. Using this approach, we were able to identify the region within a gene that gives the largest log-fold change values between the wild-type strain and the polarity mutant. However, we note that while using a smaller window to compare genes allows a more accurate quantification of the log-fold change, it also increases the sensitivity to outliers. When comparing the maximum log-fold changes in genes of the three modules described above, we observed no significant ($P < 0.05$, $t$ test) changes between the means of their distributions (Fig. 5C). Therefore, each module appears to be of equal importance for the functioning of the latent polarity pathway.

Although our results do not provide a complete molecular mechanism for the latent polarity pathway, they do offer important insights into its structure. The increased dependency on spatial cues we observe agrees with previous studies that have shown that the deletion of Bem1 is not lethal, but only if the bud scar remains intact (Irazoqui et al, 2003; Kozubowski et al, 2008). This

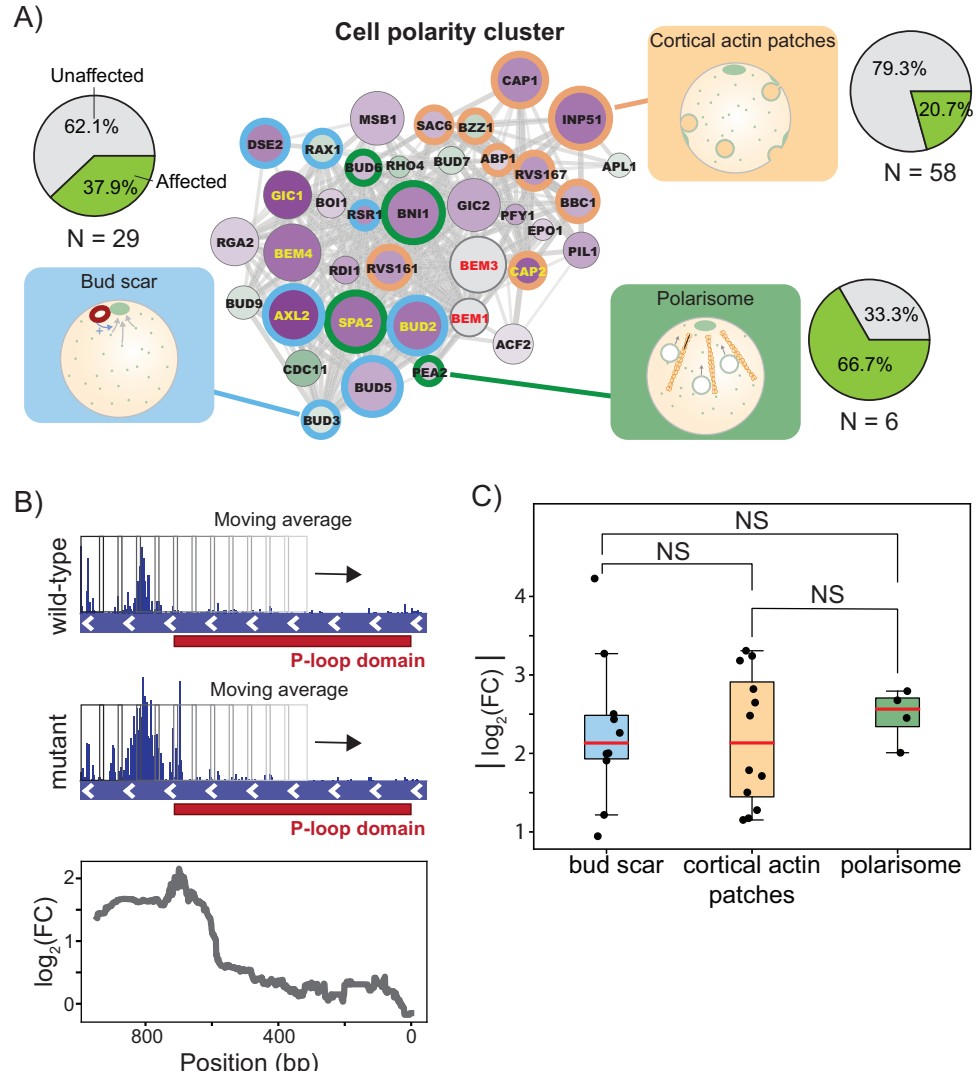

**Figure 5. Redundant polarity modules contribute to the latent polarity mechanism in a non-hierarchical manner.**

(A) Genes in the cluster enriched cell polarity factors are annotated based on whether they belong to the bud scar (blue), cortical actin patches (yellow) or polarisome (green). The fraction of all *N* genes annotated with each of these three cellular component GO terms that are present in our dataset is shown in the pie charts on the right. For all three components, at least 20% of the annotated genes are present in our dataset. The cortical actin patch has the lowest representation but is also the least polarity-specific of the three cellular components. (B) An example of the sliding average procedure taken to obtain the domain with the largest log-fold change is shown for *CDC11*. In *CDC11*, the region downstream of the P-loop domain (relative to the start codon) has an increased tolerance to transposon disruptions, while the remainder of the gene shows a similar tolerance between the two strains. Using this method, we can accurately capture this domain-dependency. The largest absolute value of the log-fold change profile over a gene was used to generate and compare the distributions shown in (C). (C) Log-fold change distributions for the genes in our dataset encoding for proteins of the bud scar (n = 11 genes), cortical actin patches (n = 12 genes), and polarisome (n = 4 genes). The orange line denotes the median of the distribution, the boxes represent the interquartile range (IQR) and the whiskers extend to the most extreme data points within 1.5 times the IQR. The three distributions all had very similar medians, and the difference in their mean was not statistically significant (P > 0.05, *t* test). Source data are available online for this figure.

dependency on the bud scar therefore appears to persist even after mutations that compensate for the fitness effects caused by losing Bem1 have occurred. In contrast, the role of actin-based transport in cell polarization has been a subject of ongoing debate (Woods et al, 2016; Irazoqui et al, 2005; Wedlich-Soldner et al, 2004; Marco et al, 2007; Slaughter et al, 2009; Slaughter et al, 2013). Our results support the role of the actin network in promoting polarity establishment, at least within the latent pathway. However, the lack of hierarchy between the different modules in the latent pathway suggests that, while they contribute to polarity

establishment, each individual module's contribution is too weak to drive polarization.

While earlier studies indicated that actin-based transport can drive polarization of Cdc42 (Klünder et al, 2013), their results were criticized to be largely caused by the unintended activation of the stress response pathway. Our data provide new evidence that actin-based transport can contribute to cell polarization, but its activity is most likely too weak to achieve polarity establishment on its own. The bud scar could provide an additional activation mechanism for Cdc42 that, in conjunction with actin recycling, is able to maintain

a polarized distribution of Cdc42. Indeed, the bud scar-associated GTPase Rsr1 is capable of recruiting Cdc24, the GEF for Cdc42, to the bud site (Park et al, 1997). This recruitment would cause a locally increased rate of Cdc42 activation near the bud scar, which, combined with a possible saturation of Cdc42 GAPs caused by the loss of Bem3, could drive polarity establishment. While such a mechanism largely agrees with the findings obtained by the model of Brauns et al (2023), a key difference is that we do predict that Cdc24 is polarized prior to budding in the polarity mutant. In addition, we propose that this latent polarity pathway may also depend on protein interactions that have not been observed in wild-type cells. For example, it is unclear why Boi1, which depends on Bem1 to interact with Cdc24, has an increased importance in the polarity mutant (Fig. EV4). The same goes for several other genes that are too numerous to discuss individually. Microscopy, in vitro experiments, and molecular models will likely be needed to obtain a complete picture of the latent polarity pathway.

In conclusion, our findings demonstrate that modularity plays a minor role in shaping evolutionary outcomes. Instead, compensatory evolution drives genome-wide rearrangements of interaction networks that extend across module boundaries, potentially facilitated by protein promiscuity. Weak and transient protein interactions—recently described as the "glue that holds the cellular network together" (Hein et al, 2015)—could be of particular importance for restoring network structure following gene loss. Because such genome-wide changes affect the adaptive potential of many cellular processes, they may help explain how gene loss can promote evolutionary novelty (Farkas et al, 2022; Helsen et al, 2020; Guijarro-Clarke et al, 2020; Cañestro and Postlethwait, 2007; Murray, 2020; Hottes et al, 2013) and allow populations to escape local fitness peaks (Helsen et al, 2020). Despite the minor role of modularity, our finding that genes within the same biological process occasionally share the same direction of disruption tolerance change after compensatory evolution offers some potential for evolutionary prediction. Our results also highlight the importance of protein ensembles and weak interactions in shaping adaptive trajectories. Mapping these weak, non-specific interactions that are frequently perceived as irrelevant will be instrumental in linking biological network structure to the evolutionary potential of biological processes.

## Methods

### Reagents and tools table

| Reagent/resource | Reference or source | Identifier or catalog number |
| --- | --- | --- |
| **Experimental models** | | |
| yEK23 (*S. cerevisiae*) | Laan Lab | N/A |
| yEK19 (*S. cerevisiae*) | Laan Lab | N/A |
| **Recombinant DNA** | | |
| pBK549 | Kornmann lab | N/A |
| Plasmid pFvO042 | Laan Lab | Dataset EV3 |
| **Oligonucleotides and other sequence-based reagents** | | |
| PCR primers | This study | Table 3 |

| Reagent/resource | Reference or source | Identifier or catalog number |
| --- | --- | --- |
| **Chemicals, enzymes, and other reagents** | | |
| Yeast Nitrogen Base without Amino Acids | Sigma-Aldrich | Y0626 |
| CSM Single Drop-Out: -Ura | Formedium | DCS0161, DCS0169 |
| CSM Single Drop-Out: - Ade | Formedium | DCS0041, DCS0049 |
| Glass beads, acid-washed | Sigma-Aldrich | G8772 |
| Phenol:Chloroform:Isoamyl Alcohol 25:24:1 Saturated with 10 mM Tris, pH 8.0, 1 mM EDTA | Sigma-Aldrich | P2069 |
| Linear Acrylamide (5 mg/ml) (1 ml Tube) | Thermo Fisher Scientific | AM9520 |
| Sodium Acetate, 3 M, pH 5.2, Molecular Biology Grade | Merck | 567422 |
| RNAse A | QIAGEN | 19101 |
| DpnII | NEB | R0543L |
| NlaIII | NEB | R0125L |
| T4 DNA Ligase (5 U/µL) | Thermo Fisher Scientific | EL0011 |
| **Software** | | |
| Transposonmapper | PyPI | v1.1.4 |
| Cytoscape | cytoscape.org | Version 3.9.1 |
| BBTools | JGI.DOE.GOV | Version 39.01 |
| **Other** | | |
| Nalgene® bottle-top sterile filter units | Merck | Z358223 |
| RNase-free Microfuge Tubes | Thermo Fisher Scientific | AM12450 |
| NEB Next Ultra II DNA Library Prep Kit | NovoGene | N/A |
| Wizard® SV Gel and PCR Clean-Up System | Promega Corporation | A9281 |

## Strains

All strains used in this study are from the W303 genetic background, and their genotypes are listed in Table 1. The *bem1* Δ*bem3* Δ*nrp1* Δ strain used in this study is the reconstructed polarity mutant from Laan et al (2015). This strain was derived from a diploid by replacing one copy of the loci encoding for *BEM1*, *BEM3*, and *NRP1* with the kanMX, natMX, and hphMX cassettes, respectively. The resulting heterozygous diploid strain was sporulated on plates selecting for the kanMX, natMX, and hphMX drug markers to obtain a haploid strain in which *BEM1*, *BEM3*, and *NRP1* are knocked out. The possible consequences of the genetic construction of the polarity mutant on the SATAY screen are examined in Appendix Figs. S5 and S6, and discussed in Appendix Section C. The Cas9 cassette was obtained from plasmid p414-TEF1p-cas9-CYC1t (Dicarlo et al, 2013) and fused to the up-and downstream genomic sequences of the *HO*-locus and the *ScURA3* marker using overlap-extension PCR. The resulting genetic construct was transformed into the wild-type strain and the polarity mutant according to a lithium-acetate transformation protocol (Gietz and Schiestl, 2007). Correct integration was verified

**Table 1. Strains used in this study.**

| Background | Name | Mating-type | Genotype |
|---|---|---|---|
| Wild-type | yEK19 | *α* | can1-100 leu2-3,112 his3-11,15 ura3ˆ0<br>BUD4 from S288C ade2::ho::Cas9 BEM1 BEM2 BEM3 NRP1 |
| Polarity mutant | yEK23 | a | MFAprHIS3 @ CAN1 locus leu2-3,112<br>his3-11,15 ura3ˆ0<br>BUD4 from s288C ade2::ho::Cas9 bem1::kanMX BEM2 bem3::natMX nrp1::hphMX |

**Table 2. Liquid media used for SATAY library generation.**

| Step | Media | Components |
|---|---|---|
| Preculture | SD-Ura+0.2% Glucose +2% Raffinose | • YNB w/o Amino Acids (6.8 g/L)<br>• CSM -Ura (0.77 g/L)<br>• Glucose (2 g/L)<br>• Raffinose (20 g/L)<br>• Adenine (20 mg/L) |
| Induction | SD-Ura+2% Galactose | • YNB w/o Amino Acids (6.8 g/L)<br>• CSM -Ura (0.77 g/L)<br>• Galactose (20 g/L)<br>• Adenine (20 mg/L) |
| Reseed | SD-Ade+2% Glucose | • YNB w/o Amino Acids (6.8 g/L)<br>• CSM -Ade (0.78 g/L)<br>• Glucose (20 g/L) |

*YNB* yeast nitrogen base, *CSM* complete supplement mixture, *Ura* uracil, *Ade* adenine.

with colony PCR. Endogenous expression of Cas9 from the *HO*-locus had no significant effects on growth. Strains were made compatible with SATAY by removing the *ScURA3* marker and the endogenous *ADE2* locus using the CRISPR/Cas9 system according to the double guide-RNA (gRNA) method of Mans et al, (2015). gRNA sequences targeting the *ScURA3* marker and *ADE2* locus were designed using the online toolbox CHOPCHOP (Labun et al, 2019). The repair fragment for removal of the *ScURA3* marker was constructed by PCR amplification of the genomic sequence upstream and downstream of the *ScURA3* marker and fusing the two fragments using overlap-extension PCR. Similarly, the repair fragment for removal of the *ADE2* locus was constructed by fusing the up- and downstream genomic sequences with overlap-extension PCR. Removal of *ScURA3* marker and the *ADE2* was verified by colony PCR using primers 15 and 16 and primers 17 and 18, respectively. Correct clones were cured from their gRNA plasmids by inoculating them in non-selective medium (YPD) and growing them until saturation (~1.5 days) at 30 °C. After saturation was reached, the cultures were plated to single colonies on YPD agar plates and screened for the loss of growth on media selecting for the marker of the gRNA plasmid. Strains were stored at −80 °C as frozen stocks in 40% (v/v) glycerol. The double gRNA plasmids that were used as a template were a kind gift from Pascale Daran-Lapujade.

## Media

Media composition and preparation protocols are the same as described in Kingma et al, 2025. Standard culturing and growth assays were performed in YPD (10 g/L Yeast extract, 20 g/L Peptone, 20 g/L dextrose), SC (6.9 g/L Yeast nitrogen base, 0.75 g/L Complete supplement mixture, 20 g/L dextrose). For *ade*⁻ strains, standard growth media was supplemented with 20 mg/L adenine just before incubation. Liquid media for the preculture and induction steps of SATAY were prepared according to the recipe in Table 2. After preparation, the media were filter-sterilized using Rapid-Flow Sterile Disposable Filter Units (Nalgene) and stored at 4 °C until use. Liquid media for the reseed step of SATAY was prepared by autoclaving 2.6 L of MiliQ water in a 5-L flask. In total, 400 mL of a 7.5× concentrated solution of the nutrients was prepared separately and filter-sterilized. To prevent the degradation of media components, this concentrate was stored in the dark at 4 °C until used. On the day of reseed, the concentrate was aseptically added to the 5-L flask containing 2.6 L of MiliQ water and mixed. Solid media was prepared by adding 20 g/L agar and 30 mM Tris-HCl (pH 7.0) to the liquid media recipe and autoclaving the mixture for 20 min at 121 °C. 20 mg/L adenine was aseptically added after autoclaving, unless plates were intended to be selective for adenine auxotrophy.

## Transposon mutagenesis screens

The procedures used for the generation and analysis of the SATAY library largely follow those described by Kingma et al, 2025, with minor modifications.

### Library generation

SATAY libraries were generated based on the procedure described by (preprint: Michel et al, 2019), which is a modification of the original protocol (Michel et al, 2017) to allow transposition in liquid media. *ade*⁻ cells were transformed with plasmid pBK549 (preprint: Michel et al, 2019), which was a kind gift from Benoît Kornmann, according to a lithium acetate transformation protocol (Gietz and Schiestl, 2007). To screen for clones transformed with the intact version of plasmid pBK549 (see Michel and Kornmann (2022) for details on the different species of pBK549), 12–24 colonies were picked from the transformation plate, re-streaked on fresh SD-ADE and SD-URA plates and incubated for 3 days at 30 °C. For clones that showed full growth on SD-URA plates while producing a small number of colonies on SD-ADE plates, cells were scraped from the SD-URA plate and used to inoculate 25 mL of preculture media at an $OD_{600}$ of 0.20–0.28. Precultures were grown on an orbital platform shaker at 160 rpm, 30 °C until the $OD_{600}$ was between 5 and 7 (~20 h). The saturated precultures were used to inoculate 200 mL of induction media at an $OD_{600}$ of 0.10–0.27 and grown for 52 h to allow transposition to occur. The efficiency of transposition was monitored by plating samples of the liquid induction cultures on SD-ADE at $T = 0$ and $T = 52$ h and scoring the number of colonies on these plates after 3 days of incubation at 30 °C. After 52 h of induction, the resulting transposon mutagenesis libraries were reseeded in 3 L of reseed media, such that the $OD_{600}$ at the beginning of the reseed was 0.21–0.26. Typically, this meant that around 7 million transposon mutants were reseeded per

library. Reseeded libraries were grown for 92 h at 140 rpm, 30 °C. At the end of reseed, cells were harvested by centrifugation of the reseed cultures at 5000 × g for 30 min. Cell pellets were stored at −20 °C.

### Genomic DNA extraction

A 500 mg frozen pellet was resuspended in 500 μL cell breaking buffer (2% Triton X-100, 1% SDS, 100 mM NaCl, 100 mM Tris-HCl pH 8.0, 1 mM EDTA) and distributed into 280 μL aliquots. 300 μL of 0.4–0.6 mm glass beads (Sigma-Aldrich, G8772) and 200 μL of Phenol:Chloroform:isoamyl alcohol 25:24:1 (Sigma-Aldrich, P2069) were added to each aliquot and cells were lysed by vortexing the samples with a Vortex Genie 2 at maximum speed at 4 °C for 10 min. 200 μL of TE buffer was added to each lysate, after which the samples were centrifuged at 16,100 × g, 4 °C for 5 min. After centrifugation, the upper layer (~400 μL) was transferred to a clean Eppendorf tube. 2.5 volumes of 100% absolute ethanol were added to each sample and mixed by inversion to precipitate the genomic DNA. After precipitation, the DNA was pelleted by centrifugation at 16,100 × g, 20 °C for 5 min. The supernatant was removed, and the DNA pellet was resuspended in 200 μL of 250 μg/ml RNAse A solution (Qiagen, Cat. No. 19101). The resuspended DNA pellets were incubated at 55 °C for 15 min to allow digestion of the RNA. After digestion, 20 μL 3 M, pH 5.2 sodium acetate (Merck) and 550 μL 100% absolute ethanol were added to each sample and mixed by inversion. DNA was pelleted by centrifugation at 16,100 × g, 20 °C for 5 min. Pellets were washed with 70% absolute ethanol and dried at 37 °C for 10 min or until all ethanol had evaporated. The dried pellets were resuspended in a total volume of 100 μL MiliQ water, and the concentration of the genomic DNA samples was quantified on a 0.6% agarose gel using the Eurogentec Smartladder 200bp–10kb as a reference. Prepared DNA samples were stored at −20 °C or 4 °C until used.

### Library sequencing

To prepare genomic DNA samples for sequencing, 2 × 2 μg of DNA from each sample were transferred to non-stick microcentrifuge tubes and digested with 50 units of DpnII and NlaIII in a total volume of 50 μL for 17 h at 37 °C. After digestion, the restriction enzymes were heat-inactivated by incubating the samples at 65 °C for 20 min. Digestion results were qualitatively assessed by visualization on a 1% agarose gel stained with Sybr-Safe. Successfully digested DNA samples were circularized in the same tube using 25 Weiss units of T4 DNA ligase (Thermo Scientific, Catalog #EL0011) at 22 °C for 6 h in a total volume of 400 μL. After ligation, the circularized DNA was precipitated using 1 ml 100% absolute ethanol, 20 μL 3 M, pH 5.2 sodium acetate (Merck), and 5 μg linear acrylamide (Invitrogen, AM9520) as a carrier. DNA was precipitated for at least 2 days at −20 °C. Precipitated DNA was pelleted by centrifugation for 20 min at 16,100 × g at 4 °C and washed with 1 ml of 70% ethanol. After washing, the DNA was re-pelleted by centrifugation for 20 min at 16,100 × g at 20 °C, the supernatant was removed, and pellets were dried for 10 min a 37 °C. Each dried pellet was resuspended in water and used as a template for 20 PCR reactions of 50 μL. Transposon–genome junctions were amplified using the barcoded primers 1 and 2 (Table 3) for DpnII digested DNA or primers 3 and 4 (Table 3) for

NlaIII-digested DNA on a thermal cycler (Bio-Rad C1000 Touch). PCR amplified samples were purified using the NucleoSpin Gel and PCR cleanup kit (Macherey-Nagel) and quantified on the NanoDrop 2000 spectrophotometer (Thermo Scientific). For each sample, equal ratios (w/w) of DpnII and NlaIII-digested DNA were pooled. Library preparation and sample sequencing were performed by Novogene (UK) Company Limited. Sequencing libraries were prepared with the NEBNext Ultra II DNA Library Prep Kit, omitting the size selection and PCR enrichment steps. DNA libraries were sequenced on the Illumina NovaSeq 6000 platform using Paired-End (PE) sequencing with a read length of 150 bp.

### Sequence alignment

FASTQ files obtained from the NovaSeq 6000 platform were demultiplexed into DpnII and NlaIII-digested DNA samples based on the barcodes introduced during PCR amplification. Read pairs with non-matching barcodes were discarded. After demultiplexing, the forward read of each read pair was selected, and the sequences upstream of primer 688_minidsSEQ1210 (Michel et al, 2017) and downstream of the DpnII (GATC) or NlaIII (CATG) restriction site were trimmed. All demultiplexing and trimming steps were performed with BBduk (Bushnell et al, 2017) integrated into a home-written pipeline written in Bash. After trimming, the forward reads were aligned to the S288C reference genome (version R64-2-1_20150113) with the Transposonmapper pipeline (Iñigo de la Cruz et al, 2022 version v1.1.4) using the following settings:

- Data type: 'Single-end'
- Trimming software: 'donottrim'
- Alignment settings: '-t 1 -v 2'

## Volcano plots

Read count distributions were corrected for spikes using the Beta-Geometric correction method and subsequently normalized for differences in transposon density and sequencing depth with the median of ratios normalization (see Appendix Section A). After normalization, the total number of reads mapping to a coding sequence was summed for each replicate sample. Five reads were added as a background value to the read count of each coding sequence to allow analysis of genes where no transposon insertions had occurred. The arithmetic mean of the summed read counts from the six replicate samples was taken, and these mean values were converted into fold changes using the formula:

$$FC_G = \frac{\overline{R}_{PM,G}}{\overline{R}_{WT,G}}. \tag{1}$$

Where $FC_G$ is the fold change of gene $G$, $\overline{R}_{PM,G}$ is the mean summed read count of gene $G$ in the polarity mutant, and $\overline{R}_{WT,G}$ is the mean of the summed read count of gene $G$ in the wild-type. $P$ values were generated using the unequal variance independent $t$ test available from the SciPy library in Python (Virtanen et al, 2020) and corrected for multiple hypothesis testing with the Benjamini–Hochberg procedure implemented in the TRANSIT software tool (DeJesus et al, 2015, version 3.2.6). Genes were considered to be essential if they had been annotated as such by the Saccharomyces Genome Database (Cherry et al, 2012, accessed on 02/19/2025).

**Table 3. List of used primers.**

| # | Name | Barcode | Sequence (5'–3') |
|---|---|---|---|
| 1 | HT50_688_minidsSEQ1210 | HT50 | GCC ACA TAT TTA CCG ACC GTT ACC GAC CGT TTT CAT CCC TA |
| 2 | E2_HT48_MiniDS_RV | HT48 | AGG TCA GTC ACA TGG TTA GGA CGC AGA GCT GAA ACG AAA ACG AAC GGG ATA AA |
| 3 | HT60_688_minidsSEQ1210 | HT60 | TAG GAT GAT TTA CCG ACC GTT ACC GAC CGT TTT CAT CCC TA |
| 4 | E2_HT49_MiniDS_RV | HT49 | AGG TCA GTC ACA TGG TTA GGA CGC AGA TAG ACA ACG AAA ACG AAC GGG ATA AA |
| 5 | Axl2_upstr_1 | – | AAA CCG CCA CAC TGT CAT TAT TAT AAT TAG AAA C |
| 6 | Axl2_upstr_2 | – | gct ggc cgg gtg acc GGT GGC TGT TGC AGT GTC AAT |
| 7 | Axl2_downstr_1 | – | gcg gtg tga aat acc gca cag aGA ATA ATC AAA GGC CCC ACG TCA G |
| 8 | Axl2_downstr_2 | – | TGT GGA TAA AGA CGA TCA GAT CAT TAC GG |
| 9 | Dyn1_upstr_1 | – | CTT TCG TTC CCA CAA TTG CCA CC |
| 10 | Dyn1_upstr_2 | – | gct ggc cgg gtg acc TGC CTA AAA AAC GTT TTG ACG TAC TTT CCA A |
| 11 | Dyn1_downstr_1 | – | tgc ggt gtg aaa tac cgc aca gaC CTT TTC AGG TAC GCG TGT CTT G |
| 12 | Dyn1_downstr_2 | – | GGA AGC CAA GGC AAG GCT G |
| 13 | Ura_1 | – | TTG GAA AGT ACG TCA AAA CGT TTT TTA GGC Agg tca ccc ggc cag c |
| 14 | Ura_2 | – | CAA GAC ACG CGT ACC TGA AAA GGt ctg tgc ggt att tca cac cgc a |
| 15 | Cas9-FSeq9 | – | AAG GTA ACG AGC TGG CAC TG |
| 16 | HO-RF-RV | – | TGCCGTCGAAAAGTCTACCGG |
| 17 | Primer 10 Reverse Check | – | CATATTGGAAGACCTTCCAAGGGAACATTATAG |
| 18 | Primer 9 ADE Forward Check | – | GAAAGCTTTTGACCAGGTTATTATAAAAGAAACTTC |

## Physical interaction network analysis

Because the correlation between evolutionary dynamics and PPI topology depends on network quality, we only included interactions from the Multi-Validated dataset. The Multi-Validated (MV) dataset of physical interactions (release BIOGRID-4.4.214) was downloaded from the BioGRID in Tab 3.0 format, which contains all physical interactions that have been validated by at least two different experimental systems or publication sources. These experimental systems include high-throughput in vitro screens, and the network therefore contains a mixture of in vitro and in vivo validated interactions. Moreover, it should be noted that the constructed PPI is a static representation of a dynamic network structure. The original set of physical interactions was filtered to obtain only those that correspond to interactions between proteins of *S. cerevisiae*. Interactions inferred from Affinity Capture-RNA or Protein-RNA were excluded from the dataset as these interactions do not represent direct physical interactions between proteins. Network visualizations were made with Cytoscape (Shannon et al, 2003, version 3.9.1). Network centralities (degree, betweenness, and closeness) were calculated using the NetworkX package in Python (Hagberg et al, 2008, version 2.8.4). A visual representation of the formulas that are used to calculate these centralities is given in Fig. EV5. The probability $P(DI)_k$ that a protein with degree $k$ is Differentially Important (DI) between the polarity mutant and the wild-type strain was calculated using:

$$P(DI)_k = \frac{N_{DI}(k)}{N_{tot}(k)}. \tag{2}$$

With $N_{DI}(k)$, the number of proteins with degree $k$ for which the corresponding gene was identified to have a significantly altered transposon insertion tolerance between the two backgrounds, and $N_{tot}(k)$, the total number of proteins in the PPI network with degree $k$. Because betweenness and closeness centrality values can be any rational number (rather than only positive integers, as is the case for $k$), betweenness and closeness values were binned prior to calculating $P(DI)$. Bin size was determined with the Freedman Diaconis Estimator method implemented in the histogram_bin_edges function of the Numpy package (version 1.26.4). After binning, the $P(DI)_a$ values for the betweenness and closeness centralities were

calculated for each bin with:

$$P(DI)a = \frac{N_{DI}(a)}{N_{tot}(a)}. \qquad (3)$$

Where $N_{DI}(a)$ is the number of proteins that have a centrality value that falls within bin $a$ and that have been identified to have a differential insertion tolerance, and $N_{tot}(a)$ is the total number of proteins in the PPI network with a centrality value that falls within bin $a$.

### Functional interaction network analysis

Functional associations between gene products were retrieved from the STRING database (Szklarczyk et al, 2021) through the stringApp plugin (Doncheva et al, 2019, version 1.7.1) of Cytoscape (Shannon et al, 2003, version 3.9.1). The confidence threshold of the imported interactions from STRING was set to medium (confidence level ≤0.40). Markov Clustering (MCL) was performed with the clusterMaker2 (version 2.2) plugin of Cytoscape using the confidence level as edge weight and setting the granularity parameter to 2.5.

### Construction of *axl2Δ* and *dyn1Δ* mutants

The *AXL2* and *DYN1* deletion mutants were generated by homologous recombination in which the open reading frame of each gene was replaced by the *URA3* auxotrophic marker. The constructs for homologous recombination were assembled by overlap-extension PCR using the following procedure. For *AXL2*, upstream and downstream genomic flanks were amplified from genomic DNA using primers 5 and 6 and primers 7 and 8, respectively (Table 3). For *DYN1*, upstream and downstream flanks were amplified from the same genomic template using primers 9 and 10 and 11 and 12, respectively. For the *AXL2* construct, the two *AXL2* flanking fragments were mixed in equimolar amounts with plasmid pFVO42C (see Dataset EV3) and used for overlap-extension PCR using primers 5 and 8 (Table 3) to generate the final deletion cassette. For the *DYN1* construct, the *URA3* marker was first amplified from plasmid pFVO42C with PCR with primers 13 and 14 (Table 3). The obtained *URA3* fragment was then combined in equimolar ratio with the *DYN1* upstream and downstream fragments as a template for overlap-extension PCR using primers 9 and 12 (Table 3) to produce the *DYN1* deletion cassette. All PCR reactions contained 5% DMSO to enhance primer-annealing specificity. Assembled constructs matching the expected deletion-cassette sizes were purified from agarose gels using the Wizard® SV Gel and PCR Clean-Up System (Promega). Liquid cultures of yEK19a and yEK23a were inoculated from glycerol stocks and grown overnight at 37 °C. On the day of transformation, each culture was divided into three biological replicates (A, B, C), and each replicate was transformed with either the *AXL2* or *DYN1* deletion construct following the method of Gietz and Schiestl, 2007 (Appendix Fig. S2).

### Transformation plate analysis

Colony numbers and sizes were determined from images of *axl2Δ* and *dyn1Δ* transformation plates using ImageJ according to the following procedure. Thresholds to binarize the images were determined manually and applied equally to all plates per deletion (see Appendix Fig. S3a). The "Analyze particles" function was used to identify objects with sizes between 6 and 2481 square pixels and a circularity of >0.6. Objects that fell between these limits were considered colonies. Appendix Fig. S3b illustrates how this procedure can correctly identify colonies while mitigating noise arising from the plate edges and reflections. The same procedure was applied to the empty negative control plates for each deletion to estimate the number of erroneously detected objects.

## Data availability

The raw and processed datasets produced in this study are available in the following database: DNA-Seq: ArrayExpress E-MTAB-15098.

The source data of this paper are collected in the following database record: biostudies:S-SCDT-10_1038-S44319-026-00709-4.

## Peer review information

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

## Acknowledgements

We thank Agnès Michel and Benoît Kornmann for providing the plasmid pBK549 containing the MiniDS transposon system, their help and guidance when setting up the SATAY assays, and insightful discussions. We are grateful to Melanie Wijsman and Nicole Bennis for providing us with the plasmids for CRISPR/Cas9 gene editing and their support during troubleshooting. We also thank Leila Iñigo de la Cruz and Gregory van Beek for establishing and maintaining the SATAY data analysis pipeline and their advice regarding data management. LL and EK gratefully acknowledge funding from the European Research Council under the European Union's Horizon 2020 research and innovation programme (grant agreement 758132).

## Author contributions

**Enzo Kingma**: Conceptualization; Data curation; Formal analysis; Validation; Investigation; Visualization; Methodology; Writing—original draft; Writing—review and editing. **Marieke Glazenburg**: Formal analysis; Investigation. **Karel Olavarria**: Formal analysis; Investigation. **Liedewij Laan**: Conceptualization; Supervision; Funding acquisition; Investigation; Writing—review and editing.

Source data underlying figure panels in this paper may have individual authorship assigned. Where available, figure panel/source data authorship is listed in the following database record: biostudies:S-SCDT-10_1038-S44319-026-00709-4.

## Disclosure and competing interests statement

The authors declare no competing interests.

# Expanded View Figures

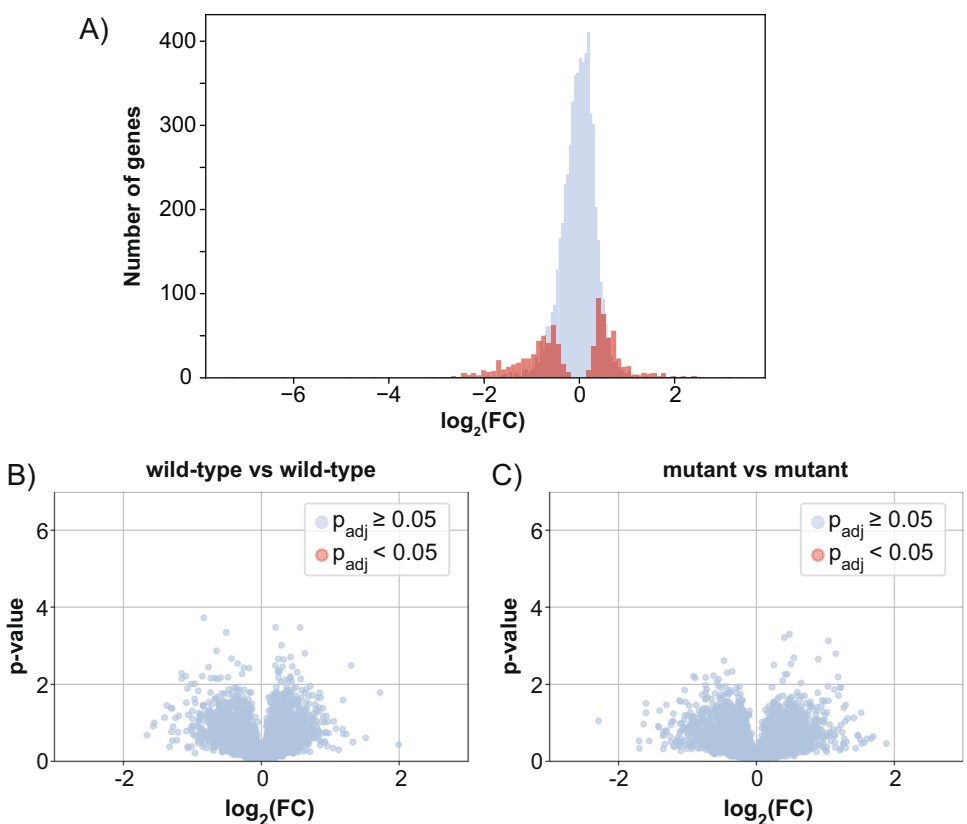

**Figure EV1.   Validity of the gene set that is identified as having differential fitness between the wild-type strain and the polarity mutant.**

(A) Significant genes have a larger effect size. Histogram of the $\log_2$-fold changes shown in Fig. 2. The plot shows that genes that are flagged to have a statistically significant difference between the two genetic backgrounds (red bars) typically have larger fold changes. (B, C) Comparing transposon mutagenesis libraries obtained from the same genetic background yields no significant differences in gene fitness. Volcano plots are shown for comparisons between wild-type and mutant datasets. The replicate datasets of each genetic background (6 in total) were split and compared 3 vs. 3. Statistical significance was determined with Welch's $t$ test and corrected for multiple hypothesis testing with the Benjamini–Hochberg procedure ($p_{adj}$). For both our wild-type strain and our polarity mutant, no false positives are found at a significance threshold of $P_{adj} < 0.05$.

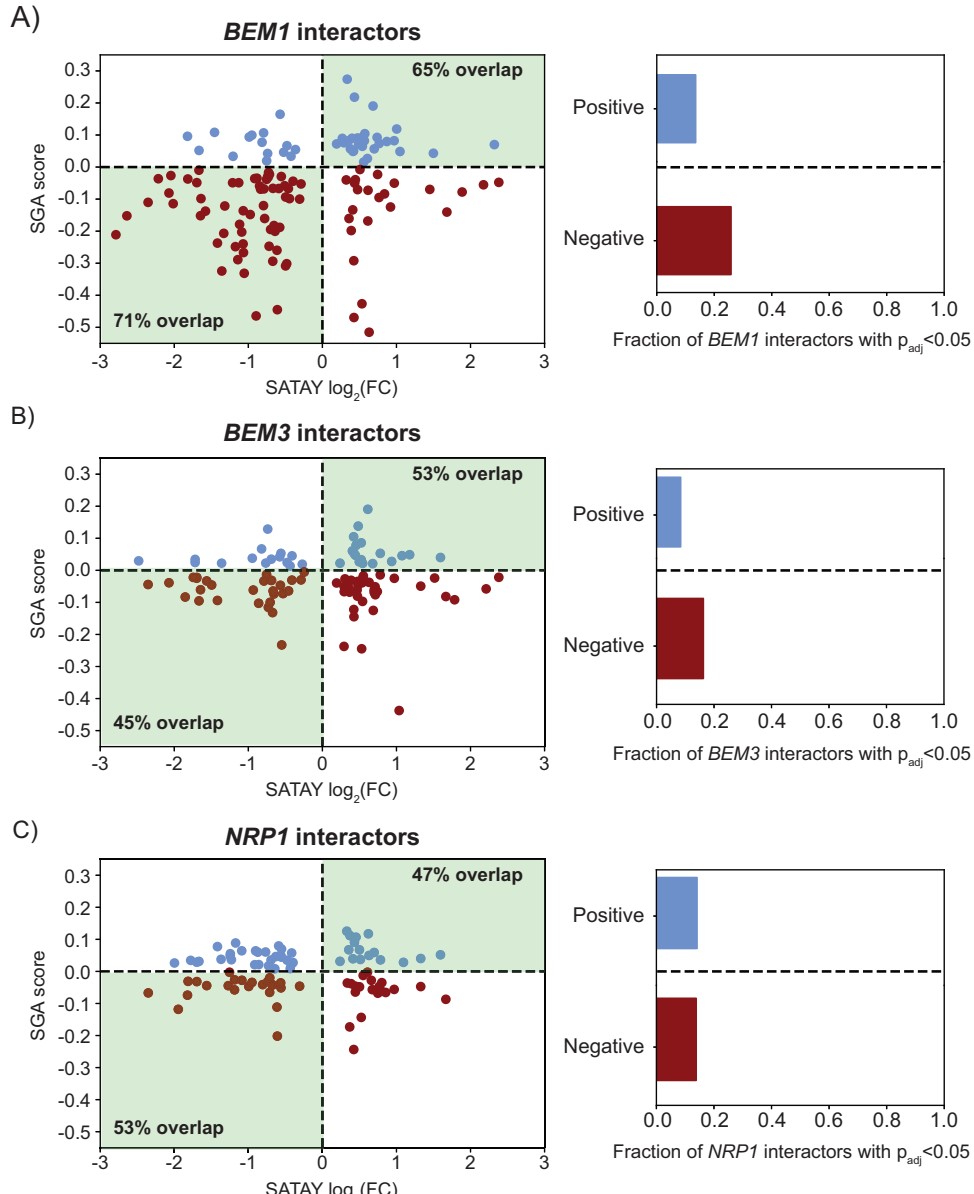

**Figure EV2.  Comparison of the genes with differential insertion tolerance found with SATAY to the genetic interactions identified by the SGA screen from Costanzo et al (2016).**

Scatterplots illustrate the SGA scores plotted against the log-fold changes in gene tolerance for genes that both exhibit differential insertion tolerance in our SATAY screen and are identified as genetic interactors of (A) *BEM1*, (B) *BEM3*, or (C) *NRP1* by the SGA screen. Positive interactors (SGA score >0) are shown in blue, while negative interactors (SGA score <0) are shown in red. The percentage of genes with matching signs between the SGA score and $\log_2$ (FC) is displayed within each scatterplot (green-shaded area). The panels to the right of each scatterplot indicate the fraction of positive and negative genetic interactors from the SGA screen that were categorized as having differential insertion tolerance by the SATAY screen.

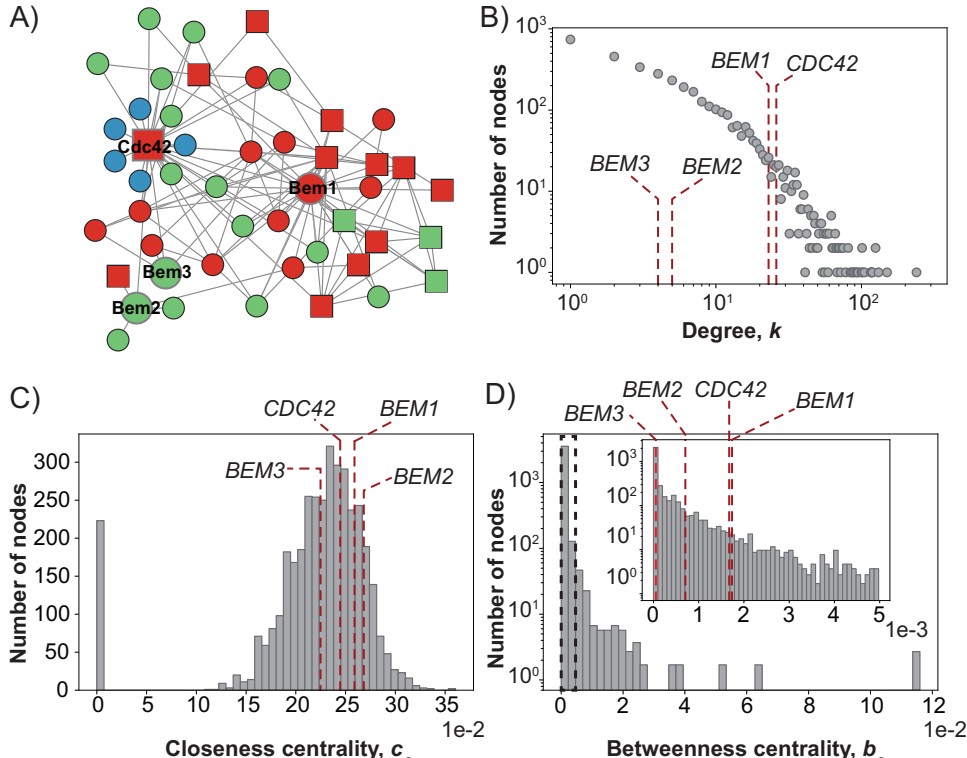

**Figure EV3.  Properties of the constructed protein-protein interaction network.**

(**A**) Sub-graph of the protein-protein interaction network for Cdc42, Bem1, Bem2 and Bem3 and their first neighbors. Nodes are colored according to their degree $k$ in the complete PPI network (Fig. 3A). Blue: $k \leq 3$, green: $4 \leq k \leq 10$, red: $k > 10$. Proteins that are essential according to the SGD database are shown as squares, non-essential proteins are shown as circles. (**B**) The degree centrality of the nodes in the complete PPI network. The degree distribution shows the typical sub-linearity of PPI networks in biology when plotted on a log-log scale. (**C**) The closeness centrality distribution of the PPI network. (**D**) the betweenness centrality distribution of the PPI network. The inset presents a zoomed-in view of the area outlined by the black dashed square. In (**C, D**), the degree values are indicated for two polarity proteins that have a strong negative effect on fitness when deleted (Bem1 and Cdc42) and for two polarity proteins that have a moderate to weak negative effect (Bem3 and Bem2). With respect to degree and betweenness, proteins with a similar gene fitness lie in proximity of each other on the distributions. For the closeness distribution, we find no relation with gene fitness.

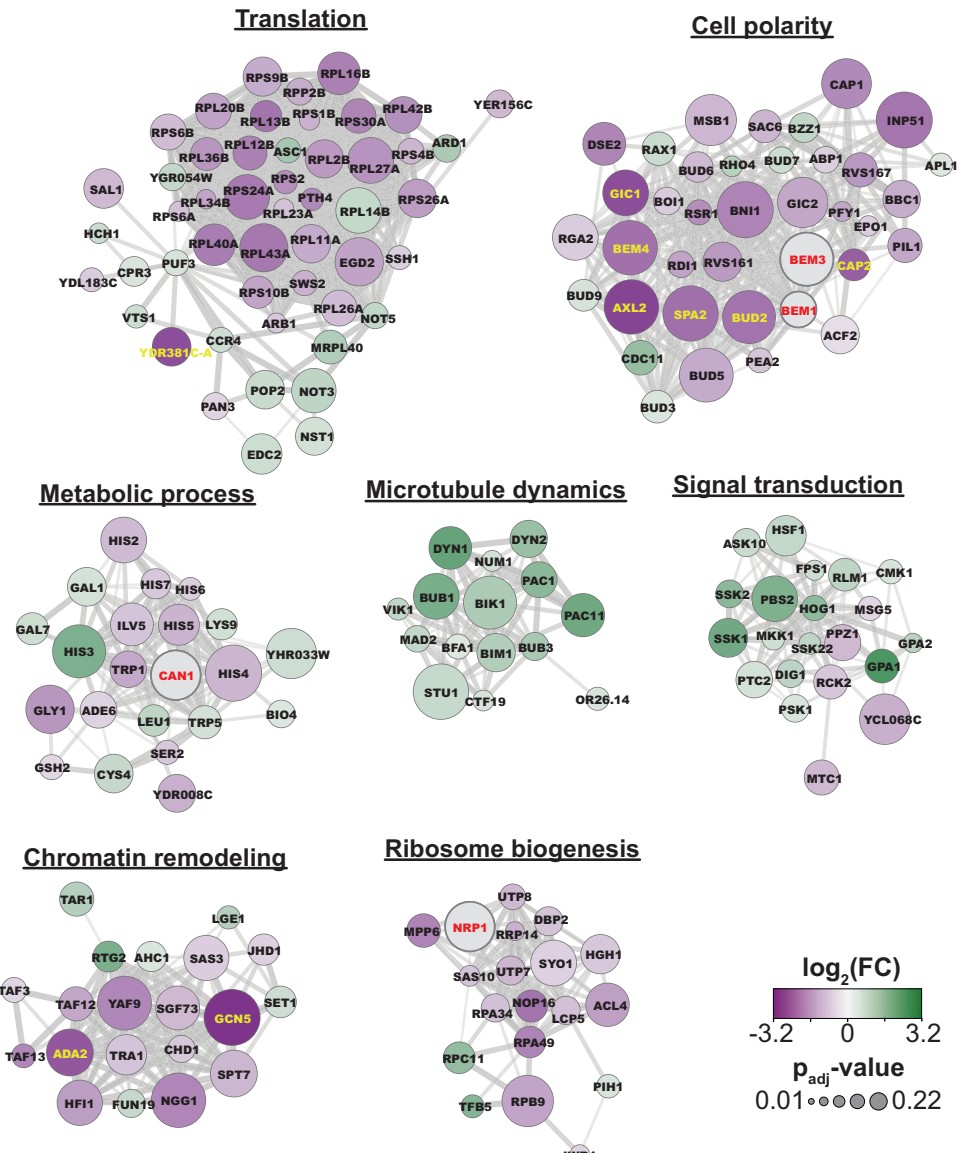

**Figure EV4. Zoom-in of the seven largest clusters identified in our functional association network by the Markov clustering algorithm.**

The biological process gene ontology enrichment is shown above each cluster.

A)

$$k(i) = \Sigma e_{ij}$$

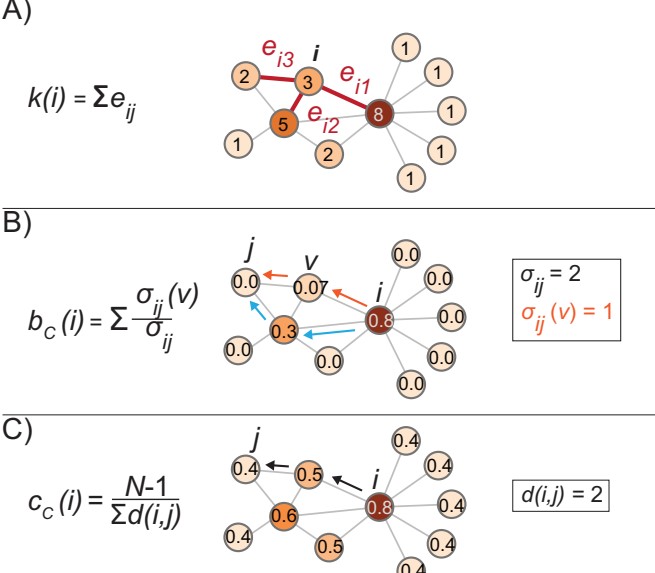

B)

$$b_C(i) = \Sigma \frac{\sigma_{ij}(v)}{\sigma_{ij}}$$

$\sigma_{ij} = 2$
$\sigma_{ij}(v) = 1$

C)

$$c_C(i) = \frac{N-1}{\Sigma d(i,j)}$$

$d(i,j) = 2$

**Figure EV5.   A visual representation illustrating how the different centrality measures are calculated.**

(A) The degree centrality $k$ is determined by counting the number of edges $e_{ij}$ a node has connecting it to other nodes in the network. The example graph shows a node $i$ with a degree of three. (B) The betweenness centrality $b_c$ provides a measure for the importance of a node for the information flow in the network based on the number of shortest paths that pass through that node. In the example there are two shortest paths $\sigma_{ij}$ from node $i$ to node $j$, but only one of these paths passes through node $v$. (C) The closeness centrality reflects the distance of a node to all other $N$ nodes in the network based on the average shortest path $d(i, j)$. The example graph shows a node $i$ with a shortest path length of two to node $j$.

