## [Peer Review File · EMBO Reports]

Global genetic rewiring during compensatory evolution in the yeast polarity network

Enzo Kingma, Marieke Glazenburg, Karel Olavarria, and Liedewij Laan

Corresponding author(s): Liedewij Laan (l.laan@tudelft.nl)

Review Timeline:

Submission Date:	2nd Oct 24
Editorial Decision:	22nd Nov 24
Revision Received:	7th Apr 25
Editorial Decision:	13th Jun 25
Revision Received:	27th Oct 25
Accepted:	18th Dec 25

Transaction Report:

Dear Dr. Laan

Thank you for the submission of your research manuscript to our journal. Three referees agreed to review your manuscript. So far, we have received two referee reports that are copied below. Given that both referees are in fair agreement that you should be given a chance to revise the manuscript, I would like to ask you to begin revising your study along the lines suggested by the referees.

Please note that this is a preliminary decision made in the interest of time, and that it is subject to change should the third referee offer very strong and convincing reasons for this. As soon as we receive the final report on your manuscript, we will forward it to you as well.

Please address all referee concerns in a complete point-by-point response. Acceptance of the manuscript will depend on a positive outcome of a second round of review. It is EMBO Reports policy to allow a single round of revision only and acceptance or rejection of the manuscript will therefore depend on the completeness of your responses included in the next, final version of the manuscript.

We realize that it is difficult to revise to a specific deadline. In the interest of protecting the conceptual advance provided by the work, we recommend a revision within 3 months (February 22nd). Please discuss the revision progress ahead of this time with the editor if you require more time to complete the revisions.

I am also happy to discuss the revision further via e-mail or a video call, if you wish.

*****IMPORTANT NOTE:

We perform an initial quality control of all revised manuscripts before re-review. Your manuscript will FAIL this control and the handling will be delayed IN CASE the following APPLIES:

- 1) A data availability section providing access to data deposited in public databases is missing. If you have not deposited any data, please add a sentence to the data availability section that explains that.
- 2) Your manuscript contains statistics and error bars based on $n=2$. Please use scatter blots in these cases. No statistics should be calculated if $n=2$.

When submitting your revised manuscript, please carefully review the instructions that follow below. Failure to include requested items will delay the evaluation of your revision.*****

- 1) a .docx formatted version of the manuscript text (including legends for main figures, EV figures and tables). Please make sure that the changes are highlighted to be clearly visible.
- 2) individual production quality figure files as .eps, .tif, .jpg (one file per figure). Please download our Figure Preparation Guidelines (figure preparation pdf) from our Author Guidelines pages <https://www.embopress.org/page/journal/14693178/authorguide> for more info on how to prepare your figures.
- 3) a .docx formatted letter INCLUDING the reviewers' reports and your detailed point-by-point responses to their comments. As part of the EMBO Press transparent editorial process, the point-by-point response is part of the Review Process File (RPF), which will be published alongside your paper.
- 4) a complete author checklist, which you can download from our author guidelines (<<https://www.embopress.org/page/journal/14693178/authorguide>>). Please insert information in the checklist that is also reflected in the manuscript. The completed author checklist will also be part of the RPF.
- 5) Please note that all corresponding authors are required to supply an ORCID ID for their name upon submission of a revised

manuscript (<<https://orcid.org/>>). Please find instructions on how to link your ORCID ID to your account in our manuscript tracking system in our Author guidelines (<<https://www.embopress.org/page/journal/14693178/authorguide#authorshipguidelines>>)

6) We replaced Supplementary Information with Expanded View (EV) Figures and Tables that are collapsible/expandable online. A maximum of 5 EV Figures can be typeset. EV Figures should be cited as 'Figure EV1, Figure EV2' etc... in the text and their respective legends should be included in the main text after the legends of regular figures.

7) Before submitting your revision, primary datasets (and computer code, where appropriate) produced in this study need to be deposited in an appropriate public database (see <<https://www.embopress.org/page/journal/14693178/authorguide#dataavailability>>).

The accession numbers and database should be listed in a formal "Data Availability " section (placed after Materials & Method) that follows the model below (see also <<https://www.embopress.org/page/journal/14693178/authorguide#dataavailability>>). Please note that the Data Availability Section is restricted to new primary data that are part of this study.

Data availability

Additional information on source data and instruction on how to label the files are available <<https://www.embopress.org/page/journal/14693178/authorguide#sourcedata>>.

10) Figure legends and data quantification:

- the name of the statistical test used to generate error bars and P values,
 - the number (n) of independent experiments (please specify technical or biological replicates) underlying each data point,
 - the nature of the bars and error bars (s.d., s.e.m.)
- If the data are obtained from n {less than or equal to} 5, show the individual data points in addition to the SD or SEM.
- If the data are obtained from n {less than or equal to} 2, use scatter blots showing the individual data points.

11) Our journal encourages inclusion of *data citations in the reference list* to directly cite datasets that were re-used and obtained from public databases. Data citations in the article text are distinct from normal bibliographical citations and should directly link to the database records from which the data can be accessed. In the main text, data citations are formatted as follows: "Data ref: Smith et al, 2001" or "Data ref: NCBI Sequence Read Archive PRJNA342805, 2017". In the Reference list, data citations must be labeled with "[DATASET]". A data reference must provide the database name, accession number/identifiers and a resolvable link to the landing page from which the data can be accessed at the end of the reference. Further instructions are available at <<https://www.embopress.org/page/journal/14693178/authorguide#referencesformat>>.

12) All Materials and Methods need to be described in the main text using our 'Structured Methods' format. According to this format, the Methods section includes a Reagents and Tools Table (listing key reagents, experimental models, software and relevant equipment and including their sources and relevant identifiers) followed by a Methods and Protocols section describing the methods, ideally using a step-by-step protocol format. The aim is to facilitate adoption of the methodologies across labs. Please download and fill our Reagents and Tools Table template (.docx), which you can find in our author guidelines:

13) As part of the EMBO publication's Transparent Editorial Process, EMBO Reports publishes online a Review Process File to accompany accepted manuscripts. This File will be published in conjunction with your paper and will include the referee reports, your point-by-point response and all pertinent correspondence relating to the manuscript.

Yours sincerely,

=====

Referee #1:

In their manuscript, Kingma and Laan use TN-seq to study the impact of gene perturbations in two genetic backgrounds that differ by the inactivation of the BEM1-BEM3-NRP1 cell polarity module. When comparing both datasets, they identify almost the same number of genes with increased or decreased fitness. They then use this data to ask fundamental questions on patterns that might characterize compensatory evolution. Of the properties they studied, only membership in a functional cluster seemed predictive of the fitness effect of mutants, with cluster members tending to have similar effects.

Overall, I enjoyed the manuscript, which I found was well written, compact, and impactful. IN terms of analysis, I think the methods the author used are appropriate to answer the questions they asked. However, I think some aspects of the model system and experimental design might need to be clarified or explored in more detail to limit potential bias. In particular:

1. From the methods section, it is not clear if the triple mutant strain is freshly reconstructed or one of the evolved lines from Laan et al 2015. Whatever the answer is, the authors should present more details on any possible additional differences between the triple mutant and the reference strain. For example, if some of the deletions were performed with an autotrophy cassette, the added gene might impact the fitness landscape. In the case of an experimentally evolved strain, any other mutation present in the genome might also act to change the effect of gene perturbations without being involved in the cell polarity module central to the study. For example, CAN1 has several synthetic lethal genetic interactions: are those overrepresented in the differential affected genes? How could this affect the author's conclusions?

2. In Laan et al, the triple mutant is described as being able to achieve ~90% the growth rate of the wild-type. While this is obviously a massive gain over the initial *bem1* mutant, it is still a significant drop in growth rate. With the current experimental design, I don't think the authors can distinguish between differential effects due to the perturbation of the cell polarity module and the slower growth rate in itself. Do you have any way to quantify this potential bias? Could it explain the genome-wide distribution of differential effects?

3. The authors show that qualitatively speaking, the number of differentially insertion tolerant genes is the same in both directions. However, looking at the scatterplot of figure 2b, the distributions of the log₂ fold-change values is asymmetrical, with a heavier tail towards negative values. Can the authors comment on this?

4. I was confused by the screening conditions described in the methods. The authors write: "resulting transposon mutagenesis libraries were reseeded in 3 litres of reseed media at an OD₆₀₀ of 0.21-0.26. Typically, this meant that around 7 million transposon mutants were reseeded per library. Reseeded libraries were grown for 92 hours". Does that mean 0.21 OD units were transferred to 3L of media, or that the 3L of media had an OD of 0.21 after reseeded? These two scenarios are very different in terms of initial population size and number of generations. Also, how long did the cultures take to saturate? Were they in the stationary phase for a long time before being collected at the end of the experiment?

5. The authors do not seem to have validated any of the differentially tolerant genes by complete deletion and phenotyping. While I understand that the manuscript focuses on broad changes in the fitness landscape, they are still working with a heavily modified strain and so validating both the presence and the strength of the effects detected by their TN-Seq assay would be a good precaution.

Other comments

6. I found the way log₂ fold-changes were described was not intuitive. In the text, the authors often refer to gene fitness instead, which is not the same and made things harder to follow for me. I would stick to insertion tolerance/sensitivity and avoid mentioning gene fitness, especially as the authors show that fold-change values vary within coding sequences and so can be determined at the domain level (figure 5b).

7. On line 442, essentiality data from SGD was accessed in 2006. Maybe it would be worthwhile to update this data, especially in light of all the new studies on differences in gene essentiality that have been published?

Referee #2:

Thank you for the opportunity to review "Global genetic rewiring during compensatory evolution in the yeast polarity network" by Kingmaa and Laan. This manuscript describes a transposon mutagenesis screen in *bem1Δ bem3Δ nrp1Δ* budding yeast strain to understand mechanisms of compensation. Authors identify bioprocesses unrelated to cell polarity such as translation being involved in compensation.

The idea to understand the genetic interaction wiring diagram of this triple mutant is interesting. The data include a number of negative results, which is important to understand the features that have been tested to explain mechanisms of compensation.

Authors state that genes belonging to the same bioprocess show a fitness change in the same direction. This is not a novel finding since it was previously described as "monochromaticity" in Costanzo et al Science 2010.

The authors only discuss the screen of their triple mutant. However, it would be informative to analyze the transposon screen on the single mutants *bem1*, *bem3*, *nrp1* and all the possible double mutants? This would tell you if the effects on fitness from the screen of the triple mutant can be explained by digenic and trigenic interactions since you are effectively looking at quad-interactions. If not perhaps, it could be accomplished in a future study but a comparison should be made to the genetic interaction study in Costanzo et al 2016 (data are easily accessible through <https://thecellmap.org/> for digenic interactions of *bem1*, *bem3* and *nrp1*)

Figure 3e shows that when betweenness increases, strains with negative fold change in fitness show higher betweenness than those with positive fold change. This should be explained in the Discussion.

Minor comments

Line 63 - *bem1Δ* is repeated. *bem1Δ bem1Δ nrp1Δ* instead should be *bem1Δ bem3Δ nrp1Δ*

Is the figure legend correct in Figure 4c? Isn't green supposed to indicate a positive LFC and purple negative?

We sincerely thank the referees for their feedback that helped us tremendously to improve our manuscript. Below we will go through the feedback one by one with our detailed response.

Referee #1

In their manuscript, Kingma and Laan use TN-seq to study the impact of gene perturbations in two genetic backgrounds that differ by the inactivation of the BEM1-BEM3-NRP1 cell polarity module. When comparing both datasets, they identify almost the same number of genes with increased or decreased fitness. They then use this data to ask fundamental questions on patterns that might characterize compensatory evolution. Of the properties they studied, only membership in a functional cluster seemed predictive of the fitness effect of mutants, with cluster members tending to have similar effects.

Overall, I enjoyed the manuscript, which I found was well written, compact, and impactful. In terms of analysis, I think the methods the author used are appropriate to answer the questions they asked. However, I think some aspects of the model system and experimental design might need to be clarified or explored in more detail to limit potential bias. In particular:

1. From the methods section, it is not clear if the triple mutant strain is freshly reconstructed or one of the evolved lines from Laan et al 2015. Whatever the answer is, the authors should present more details on any possible additional differences between the triple mutant and the reference strain. For example, if some of the deletions were performed with an autotrophy cassette, the added gene might impact the fitness landscape. In the case of an experimentally evolved strain, any other mutation present in the genome might also act to change the effect of gene perturbations without being involved in the cell polarity module central to the study. For example, CAN1 has several synthetic lethal genetic interactions: are those overrepresented in the differential affected genes? How could this affect the author's conclusions?

We thank Referee #1 for their interest in our work and careful reading of our manuscript. We agree that the details provided on how the polarity mutant was constructed were insufficient in the original version of the manuscript. We have therefore added a clarification in the methods that the polarity mutant that is used is a reconstructed triple mutant strain that was used previously by Laan et al. This mutant differs from the wild-type strain by 3 drug markers (*kanMX*, *hphMX* and *natMX*), by one auxotrophic marker (*HIS3* introduced at the *CAN1* locus) and in its mating type (**a** vs. **α**). These differences are summarized in **Table S1**. We have also included a supplemental discussion and supplemental figures **S5** and **S6** to the manuscript to address how these additional changes might impact the fitness landscape.

In short, we do not observe an overrepresentation of genetic interactors of *HIS3* in our dataset, but we do see a pattern of increased dispensability of amino acid transports and a decreased dispensability of genes involved in histidine synthesis in the polarity mutant compared to our wild-type strain. This is the pattern that we would expect from a strain that has been changed from an auxotroph to prototroph for histidine and we therefore do not observe indications that the presence of the *HIS3* marker plays any other major role in the fitness landscape other than by allowing the triple mutant to synthesize histidine. Similarly, we did not find any clear effects linked to the difference in mating type between the two strains. Nevertheless, we acknowledge that we

cannot fully exclude that the *HIS3* marker or any of the other genetic changes might have affected the global changes in the fitness landscape we have observed. However, we reason that regardless of whether the *HIS3* marker makes a significant contribution or not, our data demonstrates that constructing a mutant that shows a similar polarity phenotype to a wild-type strain drastically changes the fitness landscape.

2. In Laan et al, the triple mutant is described as being able to achieve ~90% the growth rate of the wild-type. While this is obviously a massive gain over the initial bem1 mutant, it is still a significant drop in growth rate. With the current experimental design, I don't think the authors can distinguish between differential effects due to the perturbation of the cell polarity module and the slower growth rate in itself. Do you have any way to quantify this potential bias? Could it explain the genome-wide distribution of differential effects?

It is indeed correct that the triple mutant has a growth rate that is approximately 90% of the growth rate of the wild-type strain. Thus, the compensatory gene deletions do not completely revert the fitness deficit caused by the loss of *BEM1*. In principle, this slower growth rate should not in itself lead to a bias towards under- or overestimation of the fitness effects of gene disruptions for the following reason. In the SATAY screen, the transposon mutagenesis libraries are independently generated for each genetic background, meaning that the two strains (triple mutant and wild-type) are never in direct competition with each other. As such, the read counts obtained at the end of the screen reflect the relative abundance of a transposon mutant compared to the bulk mutant population derived from the same genetic background. Thus, a gene can be more tolerant to disruptions in the triple mutant than in the wild-type, even though these disruptions would not allow the triple mutant to outgrow the wild-type strain in a direct competition.

It is possible, however, that diminishing returns epistasis—a phenomenon where the fitness advantage of a beneficial mutation diminishes as the fitness of the genetic background in which it occurs increases—is visible in our results. Specifically, diminishing returns epistasis would cause mutations that are beneficial in both strains to be identified by our screen as having a $\log_2(\text{FC}) > 0$. We do note that this effect would only be a possible explanation for the specific cases where both the $\log_2(\text{FC}) > 0$ and the gene disruptions are beneficial in both genetic backgrounds. While it is certainly possible that these cases are represented in our data, we are not convinced that it would explain the majority of the 414 genes that were found to have a $\log_2(\text{FC}) > 0$. In addition, we do not interpret diminishing returns epistasis to be a bias of our screening method, but rather a consequence of the general complexity of the global genetic interaction network.

As we agree that we cannot distinguish between these general effects and those specific to the perturbation of the polarity module, we have added the following discussion on page 13.

“Although the polarity mutant and wild-type strains are phenotypically similar, the polarity mutant exhibits a small but significant fitness disadvantage compared to the wild-type strain. Considering this fitness difference, it is possible that diminishing returns epistasis, a phenomenon where the fitness effect of beneficial mutations diminishes in fitter genetic backgrounds, contributes the number of genes identified as having differential tolerance. Specifically, diminishing returns epistasis would cause genes whose disruption confers a positive fitness effect on both strains to appear more tolerant in the polarity mutant ($\log_2(\text{FC}) > 0$). This phenomenon represents a general

effect that arises when comparing the same beneficial mutations across strains with different fitness. Consequently, not all genes identified as more tolerant in our screen necessarily reflect a difference in genetic wiring between the two strains. However, we anticipate that the subset of genes displaying increased tolerance due to diminishing returns epistasis is limited. It involves only those genes for which disruption has a positive fitness effect in both strains. This will only be a subset of the genes with a, which is also reflected by our finding that ~7% of these genes have been annotated as essential (figure 2c), making it likely that their disruption has a negative fitness effect on the wild-type strain. In addition, even in non-optimal environments there appears to be only a limited number of genes whose mutation provides a fitness benefit in a wild-type genetic background.”

3. *The authors show that qualitatively speaking, the number of differentially insertion tolerant genes is the same in both directions. However, looking at the scatterplot of figure 2b, the distributions of the log₂ fold-change values is asymmetrical, with a heavier tail towards negative values. Can the authors comment on this?*

The distribution is indeed skewed in such a way that the magnitude of log₂(FC) changes smaller than 0 is larger than for those larger than 0. While we are cautious with providing a quantitative interpretation of the log₂(FC) change values, as they do not directly translate to fitness, we have added the following explanation for this feature of the distribution of log₂ fold changes on line 115-122:

“Interestingly, the average log-fold change in disruption tolerance is greater (p-value = 9.7e-29, Wilcoxon rank-sum test) for genes with decreased tolerance (mean = -1) than for those with increased tolerance (mean = 0.66). This asymmetry in the fold-change distribution (Figure 2a) suggests that, while the polarity mutant may not rely on more genes for fitness compared to the wild-type strain, the fitness impact of losing a gene on which it has become dependent after compensatory evolution is more pronounced. However, since the magnitude of the fold-change also depends on the size of the affected region relative to the total gene length (also see section 3), another possibility is that genes with increased disruption tolerance more often encode for multi-domain proteins in which only one domain exhibits a variation in disruption tolerance”

4. *I was confused by the screening conditions described in the methods. The authors write: "resulting transposon mutagenesis libraries were reseeded in 3 litres of reseed media at an OD600 of 0.21-0.26. Typically, this meant that around 7 million transposon mutants were reseeded per library. Reseeded libraries were grown for 92 hours". Does that mean 0.21 OD units were transferred to 3L of media, or that the 3L of media had an OD of 0.21 after reseeded? These two scenarios are very different in terms of initial population size and number of generations. Also, how long did the cultures take to saturate? Were they in the stationary phase for a long time before being collected at the end of the experiment?*

We have clarified in the methods section that the samples were reseeded such that the density of the culture was 0.21 OD at the start of the reseed by adding the following on line 418:

“such that the OD600 at the beginning of the reseed was 0.21-0.2.”

The samples will have been in stationary phase for a considerable time (>10 hours) before being collected at the end of the reseed. This long incubation allows all the different mutants present in

the SATAY libraries to reach their maximal abundance. Because the cells are not regrown after the reseed phase is finished, this should not affect the results of the SATAY screen.

5. The authors do not seem to have validated any of the differentially tolerant genes by complete deletion and phenotyping. While I understand that the manuscript focuses on broad changes in the fitness landscape, they are still working with a heavily modified strain and so validating both the presence and the strength of the effects detected by their TN-Seq assay would be a good precaution.

While we agree that validating some key genes would be very interesting and could provide useful new insights, this is something that we are planning to do as future work for the following reasons:

- We consider it a nice but not absolutely necessary control, because we did not develop the TN-seq technique here. It is an established technique to measure the effects of gene disruptions and has been extensively tested by the Kornmann lab (that developed the technique) and by other groups. See for example: Rewiring phospholipid biosynthesis reveals resilience to membrane perturbations and uncovers regulators of lipid homeostasis | The EMBO Journal, Functional mapping of yeast genomes by saturated transposition | eLife
- As the referee also says, we focus on the global picture and therefore in this paper we do not want to focus in on specific genes.
- Doing the suggested experiments in a quantitative manner is not straightforward. For example, it will require extensive strain construction to inactivate genes with effects strong enough to detect them, while simultaneously avoiding suppressor mutations and thus is beyond the scope of this paper.

6. I found the way log₂ fold-changes were described was not intuitive. In the text, the authors often refer to gene fitness instead, which is not the same and made things harder to follow for me. I would stick to insertion tolerance/sensitivity and avoid mentioning gene fitness, especially as the authors show that fold-change values vary within coding sequences and so can be determined at the domain level (figure 5b).

We chose to use gene fitness for ease of reading, but insertion tolerance is indeed the actual correct term. Because we want to avoid confusion and possible misinterpretation that the log₂ fold changes can be directly translated into gene fitness we have reverted the cases where we use the term gene fitness back to insertion tolerance/sensitivity, as suggested by Referee #1

7. On line 442, essentiality data from SGD was accessed in 2006. Maybe it would be worthwhile to update this data, especially in light of all the new studies on differences in gene essentiality that have been published?

We thank the reviewer for bringing the outdated list to our attention and have retrieved a list of essential genes from SGD (genes for which the null, conditional or dominant negative mutants are scored as inviable) from the 19th of February 2025. This slightly increased the number of genes with a log₂(FC)<0 that are annotated as essential from ~10.5% to ~11.3%. The number of genes

with a $\log_2(\text{FC}) > 0$ that are annotated as essential remained at $\sim 7.2\%$. We have updated the barplot in **panel c of Figure 2** with these new values. We have also updated the accession date mentioned in the methods section.

Referee #2

*Thank you for the opportunity to review "Global genetic rewiring during compensatory evolution in the yeast polarity network" by Kingma and Laan. This manuscript describes a transposon mutagenesis screen in *bem1Δ bem3Δ nrp1Δ* budding yeast strain to understand mechanisms of compensation. Authors identify bioprocesses unrelated to cell polarity such as translation being involved in compensation.*

The idea to understand the genetic interaction wiring diagram of this triple mutant is interesting. The data include a number of negative results, which is important to understand the features that have been tested to explain mechanisms of compensation.

Authors state that genes belonging to the same bioprocess show a fitness change in the same direction. This is not a novel finding since it was previously described as "monochromaticity" in Costanzo et al Science 2010.

We thank referee #2 for careful reading of our manuscript. It is correct that this is not a novel finding, We have added a reference to the work of Costanzo et al. to the results section, by adding the following part to the sentence on line 204-206:

"This uniform response of genes regulating the same process resembles the observed monochromatic behaviour of genetic interactions between modules of the yeast metabolic network [Segre et al.] and more generally between genes involved in the same pathway and complexes [Costanzo et al.]"

*The authors only discuss the screen of their triple mutant. However, it would be informative to analyze the transposon screen on the single mutants *bem1*, *bem3*, *nrp1* and all the possible double mutants? This would tell you if the effects on fitness from the screen of the triple mutant can be explained by digenic and trigenic interactions since you are effectively looking at quad-interactions. If not perhaps, it could be accomplished in a future study but a comparison should be made to the genetic interaction study in Costanzo et al 2016 (data are easily accessible through <https://thecellmap.org/> for digenic interactions of *bem1*, *bem3* and *nrp1*).*

Because we currently do not have SATAY data of sufficient quality on the *bem1*, *bem3* and *nrp1* mutants and because gathering this data is not trivial for mutants with low fitness such as *bem1*, we have added a comparison to the digenic interactions of these genes that are available from TheCellMap. We added the following to the discussion on line 305:

*"An important question is to what degree the individual compensatory mutations drive the global genetic rewiring that we observe. For instance, the deletion of Bem3, which plays a crucial role in polarity establishment, may have a localized impact on the fitness landscape, whereas the loss of the RNA-binding protein Nrp1 could trigger more extensive genetic changes on a global scale. A comparison with the genetic interactions mapped by a genome-wide SGA screen reveals that only a small subset (<30%) of the digenic interactions involving *BEM1*, *BEM3*, and *NRP1* appear as differentially tolerant in our SATAY dataset (figure S8). This finding suggests the prevalence of higher-order epistasis during compensatory evolution and indicates that global rewiring is not driven by the additive effects of digenic interactions. Notably, while the sign of the genetic interaction for *BEM3* and *NRP1* show limited agreement between the SGA screen and SATAY*

(figure S8b and c), those for *BEM1* are relatively well-preserved (figure S8a). This suggests that the genetic interactions associated with the strong initial perturbation are better retained compared to those of compensatory mutations.”

Figure 3e shows that when betweenness increases, strains with negative fold change in fitness show higher betweenness than those with positive fold change. This should be explained in the Discussion.

While it is correct the betweenness of genes with a negative fold change does rise above that of genes with a positive fold change as betweenness increases, we are not fully convinced that this trend is meaningful. There are relatively few genes with a high betweenness (see supplementary Figure S4d) which results in considerable fluctuations in the value of P(DI). These fluctuations make it difficult to determine whether the higher betweenness of genes with positive fold-change is a real trend or due low sampling noise. We therefore decided not to explain these differences in the discussion.

Minor comments

Line 63 – bem1Δ is repeated. Bem1Δ bem1Δ nrp1Δ instead should be bem1Δ bem3Δ nrp1Δ

We thank the reviewer for notifying us of this mistake, we have corrected it to the correct genotype *bem1Δ bem3Δ nrp1Δ*.

Is the figure legend correct in Figure 4c? Isn't green supposed to indicate a positive LFC and purple negative?

The colours have indeed been switched in the legend of figure 4C. We have modified the legend of panel C of figure 4 such that the sign of the $\log_2(\text{FC})$ matches the correct colour.

Referee #3

The manuscript "Global genetic rewiring during compensatory evolution in the yeast polarity network" by Kingma and Laan focuses on the changes in the fitness landscape that occur after compensatory evolution restores cell polarity in budding yeast following a severe genetic perturbation. The authors subjected wild-type (WT) and evolved strains to a transposon insertion library and competed the resulting clones to derive a genome-wide fitness landscape. Finally, they analyzed how the genes showing significant fitness differences between WT and evolved strains are distributed across protein interaction and functional networks.

The remarkable ability of cells to restore functionality through compensatory evolution has recently emerged as an important factor in the evolution of cell biology. However, little is still known about what dictates the evolutionary strategies and what consequences they have for evolved and compensated cells. This work makes an important contribution to elucidating the latter. Of particular relevance is the high number of genes with altered fitness compared to WT, highlighting the global impact of compensatory evolution on the fitness landscape.

Comments:

1. Since the variation in the transposon insertion analysis introduced by the authors is fairly new (Supplementary section A), validating the fold change in fitness for some key genes would strengthen the study. Deleting a couple genes with opposite effect on fitness, from the cell polarity and

microtubule networks (or others, depending on the authors' preference) in both WT and compensated strains could strengthen the approach and reveal interesting phenotypes useful for discussion.

We kindly thank referee #3 for the interest in our work. For the answer to this question we like to refer to referee1 comment 5.

2. With the current dataset, would it be possible to estimate which genes are essential in each background? For example, by identifying genes where no transposon insertions occur, accounting for a certain level of sequencing/analysis noise? I couldn't determine from the Materials and Methods how such genes are handled during analysis. If feasible, a dedicated analysis of the differential essentiality spectrum could be highly informative. In particular, identifying genes that have become essential in the compensated strain may shed light on the alternative polarization mechanisms employed by compensated cells. If not possible, a few lines to explain why is so could be useful.

We have added the following clarification to the methods section on how genes where no transposon insertions have occurred are handled during the analysis:

“Five reads were added as a background value to read count of each coding sequence to allow analysis of genes where no transposon insertions had occurred.”

In principle, genes that are conditionally essential between the two genetic backgrounds could be interpreted to exhibit a strong form of epistasis. The main issue that arises when trying analyse essential genes is the division by zero that we would encounter when computing the log-fold changes. We have taken the approach to add a background value to the read counts of all genes, which then allows us to also process the genes where no insertions have occurred in one or both of the strains. As a result, most of the genes that switch from essential to non-essential between the two genetic backgrounds should be enclosed within the genes that were identified to have a significantly difference in their tolerance to transposon disruptions. The exceptions are those genes in which only a small domain becomes essential. If this domain is too small (relative to the total gene size), and therefore contributes relatively little to the total read sum of the gene, it has a higher likelihood to be missed by our analysis method. However, this is a general problem that also affects genes that are not conditionally essential (as mentioned in the discussion).

While a more sensitive analysis of essential genes based on sub-domains within genes is possible and possibly interesting, it is also more complex and has several challenges as it requires a comparison of different genes within the same genome rather than the same gene in different genomes. For example, the analysis would then preferably need to be able to correct for differences in gene size and in the position of the gene within the genome. We therefore decided to treat essential genes the same as other genes, meaning that they also have the same limitations with respect to the level at which we can detect that they have a differential tolerance between the compensated and wild-type strains.

3. Similarly, it would be fascinating to identify (potential) genes that switch their fitness effect in compensated strains—for instance, genes whose deletion is detrimental in WT but beneficial in compensated strains, or vice versa. In theory, within each clonal competition, it should be possible to assess which genes exhibit higher or lower fitness compared to a neutral locus (e.g., HO).

We agree that identifying genes that switch their fitness effect in the compensated strain from detrimental to beneficial or vice versa would be very interesting. In principle individual mutant

fitness can be derived from the read counts obtained for the different transposon insertion sites within a gene. However, similar to the dedicated analysis for essential genes that was mentioned under comment 2, obtaining fitness values involves comparing genes that have different sizes and positions within the genome. These factors can confound the results. For instance, it has been shown that the insertion probability of the MiniDS transposon used in SATAY is influenced by its distance from the centromere (see, for example, Functional mapping of yeast genomes by saturated transposition | eLife). Consequently, comparing genes located at different distances from the centromere is not straightforward. Thus, while such an analysis would reveal interesting and relevant information, doing this properly requires the development of a more in-depth analysis method that we consider to be beyond the scope of this paper.

Minor Comments:

- *Some supplementary figure panels are not referenced in the text. The ones I identified are: S1c, S2a, S2b, S2c, S3a, S3d.*

We have added references to those panels that were not referenced to in the main text and verified that the others have been referenced.

- *It would be helpful to include more details in the Materials and Methods on how network centralities and associated likelihoods with altered fitness were calculated.*

We have added the formula's used to calculate the centralities to the **Methods** section and have also added supplemental figure **S3** as an additional reference.

- *While Figure 3A presents all 3,799 protein interactions retrieved, Figure 4A portrays only the 883 genes with differential fitness. This distinction could be highlighted more clearly to avoid misleading comparisons of the network structure.*

We have added an extra header to the panels of the two figures to make the distinction between them.

- *The color palette in Figures 3A/S3A and 3B could be misleading. Both use blue and red, but the colors represent different properties. Additionally, green is absent in Figure 3B.*

Our intention was to have the same property (protein degree) described by these colors in both figures in the same manner. In both figures, blue and red are meant to indicate the same degree values (blue: $k < 3$, red $k > 10$) of the proteins in the network shown in figure 3A.

It is not immediately clear for us why Referee #3 expects the color green to appear in figure 3B, as the colors used in that panel are intended to indicate essentiality rather than degree. We would be grateful if Referee #3 could further specify why the color green was expected to appear in panel 3B as this would help with the clarity of the manuscript.

- *The degree of connectivity expressed as k is implied but not explicitly defined.*

We have added the mathematical definition of k that we used to the **Methods** section and have updated the following sentence online 162 to have a more explicit definition of k :

"For example, hub proteins at the center of a module that interact with many other proteins have a high degree"

Dear Dr. Laan

Thank you for the submission of your revised manuscript to EMBO reports. We have now received the full set of referee reports that is copied below.

As you will see, all referees are positive about the study and support publication. That said, referee 1 and 3 remain concerned about the absence of validation experiments. Referee 3 provided further feedback on this point and noted that "extra validation is not strictly required for publication" but also noted that "I simply think that the validation of some (a couple) of the striking results from the transposon screen could strengthen the confidence of the reader". Taking these comments into account, I would encourage to perform at least some validation experiments, if this is possible. I am happy to discuss this further.

From the editorial side, there are also a few things that we need before we can proceed with the official acceptance of your study.

- Your manuscript will be published in our Reports section. To comply with this format, I kindly ask you to combine the Results and Discussion section.

- The manuscript sections should be in the following order: Title page - Abstract & Keywords - Introduction - Results and Discussion - Methods - Data Availability - Acknowledgments - Disclosure Statement & Competing Interests - References - Figure Legends - (Main Tables with legends if applicable) - Expanded View Figure Legends.

- Please remove the figures from the manuscript file.

- Please provide up to 5 keywords.

- Please provide the specific URL for the E-MTAB-15098 in the Data Availability section, once it is available.

- Please add a 'Disclosure and competing interests statement'. For more information see <https://www.embopress.org/page/journal/14693178/authorguide#conflictsofinterest>

- The references need to be alphabetical, not numerical; et al needs to be used after 10 author names; DOIs should only be used for preprints and datasets that have not been published yet (see also next point).

- Citations of preprints (e.g., Michel et al 2019) should be formatted the following way:

In the text: (preprint: NAME1 et al, YEAR);

In the reference list: Author NAME1, Author NAME2 (YEAR) article title. bioRxiv doi [PREPRINT].

- Please add callouts for Figure 1bc, Figure 4ac, Figure 5c.

- The "Supplement(al)" nomenclature should no longer be used. The Supplementary figures should either be called Figure EV# or be combined in one Appendix PDF. For EV figures, the legends are part of the manuscript in a separate section called "Expanded View Figure legends" (after the main figure legends). The EV figures are uploaded as separate, high resolution figure files. If you choose to compile them in an Appendix instead, then this needs to be one PDF containing the figures and their legends, plus a table of content with page numbers on the first page. The nomenclature in this case is Appendix Figure S1, etc. Appendix Table S1, etc. All figure callouts need to be updated accordingly.

- The source data provided for Figure 2B and 2C could also be provided as a Dataset, if that makes sense. The dataset would be called Dataset EV3 and consist of an .xls file with a legend in a separate tab.

- The text in the methods sections B - C appears to be very similar to that used in a previous publication (Kingma et al, PLOS ONE 2025). This is in principle fine (no need to reinvent the wheel for methods), but please make sure to cite this paper in the methods section.

- Please provide a Reagents and Tools Table. You can find the template for download in our author guidelines:

(Table S1 could be part of the Reagents and Tools table).

- Our production/data editors have asked you to clarify several points in the figure legends (see below). Please incorporate these

changes in the manuscript and return the revised file with tracked changes with your final manuscript submission.

A) Statistical test information. Only p-values that are actually shown in the figure panel(s) should (and must) be defined in the legends, all others should be removed from (or added to) the legend. Moreover, we ask for the specification of exact p-values:

- Please note that the exact p values are not provided in the legend of figure 4C
- Please indicate the statistical test used for data analysis in the legends of figures 2B, C

B) Replicates and error bars:

- Please note that the box plots need to be defined in terms of minima, maxima, percentile in the legend of figure 5C
- Please note that information related to n is missing in the legends of figures 2B, 5C

- Finally, EMBO Reports papers are accompanied online by

A) a short (1-2 sentences) summary of the findings and their significance,

B) 2-3 bullet points highlighting key results and

C) a schematic summary figure that provides a sketch of the major findings (not a data image).

Please provide the summary figure as a separate file in PNG or JPG format at a size of 550x300-600 pixels (width x height).

Please note that the size is rather small and that text needs to be readable at the final size. Please send us this information along with the revised manuscript.

With kind regards,

Martina

=====
Referee #1:

I have read the revised manuscript, and I found that the authors addressed most of the points raised in my review. Some comments:

Regarding point 1: I'm afraid I was unclear in the question I was trying to raise regarding the lower fitness of the triple mutant: I am aware the TN-seq screen described does not involve direct competition between the WT and mutant background. What I was alluding to is instead what fraction of changes in the fitness landscape are shaped specifically by the compensatory mechanism that restores cell polarity versus those that are instead largely driven by non-specific interactions with the reduced growth rate of the triple mutant. I think this could be discussed in the manuscript.

Regarding point 5 (and as raised by another reviewer): I have several objections to the points raised by the authors:

1) Arguing that large-scale results do not need to be validated with smaller-scale or orthogonal assays because other did it in another context is does not convince me. The authors are working in a triple mutant strain with reduced fitness and a number of peculiarities, which should motivate them to make sure they are not missing anything unexpected.

2) Validating some hits with spot assays or growth curves would not result in the paper giving the impression of focusing on a few genes.

3) I do not think performing validations would have been overly long or complex, even in the triple mutant background, especially since their strains already have Cas9 integrated at the HO locus for markerless editing and or marker recycling. Multiple synthetic inducible promoter systems exist in yeast, giving the option to build conditional expression mutants to avoid any lethal effects when deleting genes.

In short, I understand that there can be constraints on performing additional lab work, but in this context, I think this was a reasonable ask and that the way the authors justified its absence was not convincing.

Referee #2:

The authors sufficiently addressed most of the reviewer's comments.

Referee #3:

I have re-read the entire manuscript, including the comments from the other reviewers and the authors' responses. I believe the revisions have improved the manuscript.

My remaining comments to the authors are as follows:

- I agree that there are valid arguments for not re-validating an established technique. To clarify, my point was not to question the reliability of the data, but rather to highlight the added value of manually inspecting some candidates with large fold changes. I understand that the authors plan to pursue this in future work.
- Similarly, I understand the challenges posed by differences in gene length and genomic location when estimating fitness and essentiality. That said, I still believe that, while imperfect, certain approaches could yield useful information for the authors. If I am correctly understanding the protocol, sequencing the libraries prior to the fitness assay and applying the sliding window approach (as used elsewhere in the manuscript) could help identify genes with at least one essential domain, defined, for instance, as regions of at least 300 bp without insertions, accounting for background noise. This strategy would also allow quantification of barcode abundance before and after competition. Averaging the trajectories of all barcodes associated with a given gene could provide an approximate measure of fitness. Although certain genes would naturally have fewer associated barcodes and thus less precise estimates, genes showing strong shifts in fitness could still be identified. While these analyses may not be essential to support the manuscript's main conclusions, they could offer valuable additional insights.
- Lastly, I apologize for not raising this point in my initial review. I only now realized that NRP1 is a relatively uncharacterized protein, potentially RNA-binding and involved in ribosome biogenesis. This raises the question of whether the observed genetic rewiring, interpreted as global, might instead reflect the pleiotropic effects of this particular mutation, rather than being restricted to the polarity module. A brief comment from the authors on this possibility could be helpful.

Dear Martina,

Hereby our revised manuscript where we performed additional experiments to directly test the Satay predictions as we discussed via email. Our findings are described in appendix B.

In addition we ensured that our manuscript is written according to your editing guidelines.

Best regards,

Liedewij

Dr. Liedewij Laan
Delft University of Technology
Bionanoscience Department
Netherlands

Dear Liedewij,

Thank you for your patience while we have reviewed all files from the editorial side. I am very pleased to accept your manuscript for publication in the next available issue of EMBO reports. Thank you for your contribution to our journal.

You may qualify for financial assistance for your publication charges - either via a Springer Nature fully open access agreement or an EMBO initiative. Check your eligibility: <https://link.springer.com/journal/44319/how-to-publish-with-us>

Kind regards,

Martina

>>> Please note that it is EMBO Reports policy for the transcript of the editorial process (containing referee reports and your response letter) to be published as an online supplement to each paper. If you do NOT want this, you will need to inform the Editorial Office via email immediately. More information is available here: <https://link.springer.com/partners/embo-press/editorial-policies#Peer%20review>